# Enhanced multi-task learning of imputation and prediction via feature relationship graph learning

## Abstract

Missing values present significant challenges in machine learning, often degrading predictive performance. Traditional and deep learning imputation methods often overlook the relationships between features and their connections to downstream tasks. To address these gaps, we propose PIG (multi-task learning of Prediction and Imputation via feature-relationship Graph learning), a model that integrates imputation and prediction by leveraging feature interdependencies. PIG utilizes a graph-based approach to capture intricate feature relationships, thereby enhancing the accuracy of both imputation and downstream tasks. Our strategic training process begins with pre-training for both tasks, ensuring the model learns effective representations. This is followed by fine-tuning the entire model to further optimize imputation and downstream tasks simultaneously. We evaluated our method using nine benchmark datasets, three for regression and six for classification. Our method showed superior imputation and prediction performance across nine datasets, achieving an average rank of 1.33 for both imputation and regression tasks and 1.83 for imputation and 1.17 for classification tasks. Additionally, in sensitivity analysis with respect to missing rates, our method demonstrated its robustness, especially in predictive performance, compared to other methods that showed significant degradation.

## 1 Introduction

Missing values arise from various factors across diverse fields (Kim et al., 2018; Yoon et al., 2016; Li et al., 2021; Koren et al., 2009), underscoring the importance of effective imputation. Traditionally, basic statistical methods, such as mean, median, or mode, have been used for imputation; however, these approaches often yield inaccurate results. Matrix completion methods, commonly employed in recommendation systems, have demonstrated significant improvements in missing data imputation (MDI) (Koren et al., 2009; Hu et al., 2008; Rendle et al., 2014; Mazumder et al., 2010), yet they have limitations in handling categorical features. Recently, deep learning methods have been applied to MDI using AutoEncoders (AEs) and their variants (Gondara & Wang, 2018; Vincent et al., 2008), enabling flexible processing of mixed-type data that includes both numeric and categorical features. Furthermore, some studies have utilized generative models, such as Variational AutoEncoders (VAEs) and Generative Adversarial Networks (GANs), to capture the underlying data distribution and thereby generate plausible values for missing data (Nazabal et al., 2020; Yoon et al., 2018).

Despite the advancements offered by the aforementioned methods, they exhibit notable limitations. First, all these approaches focus solely on the imputation task, neglecting the downstream tasks, which are the ultimate goal of the analysis. This oversight can result in sub-optimal performance in downstream tasks. Second, while deep learning methods have demonstrated competitive accuracy in imputation, they are often sensitive to irrelevant features that are commonly present in real-world datasets (Grinsztajn et al., 2022). This sensitivity can lead to overfitting to noisy patterns of irrelevant features, hindering the generalization of imputation performance. Consequently, these limitations highlight a critical gap in both imputation and downstream tasks, necessitating the development of more integrated approaches.

More recent methods, such as GRAPE (You et al., 2020), have demonstrated the potential of Multi-Task Learning (MTL) to address both imputation and downstream tasks. Additionally, by transforming tabular data into a bipartite graph structure and using Graph Neural Networks (GNNs), GRAPE implicitly captures feature relationships through edge-level prediction tasks for imputation, while downstream tasks are framed as node-level predictions. However, despite its strong performance, GRAPE still has limitations, such as an indirect representation of feature relationships through partially observed data and challenges related to stability during MTL (Liu et al., 2021; Yu et al., 2020; Chen et al., 2018).

To this end, we propose PIG, multi-task learning of **P**rediction and **I**mputation via feature-relationship **G**raph learning. PIG incorporates a Graph Learning Layer (GLL) that explicitly learns feature relationships, which are then fully utilized in a Graph Residual Imputation Layer (GRIL) to consider only relevant features in imputation and thereby enhance its performance. Additionally, PIG enhances the stability of MTL through a carefully designed training strategy. The training process begins with the pre-training of the GLL and a Prediction Layer (PL), which is designed to perform downstream tasks. This is followed by fine-tuning the entire model to simultaneously optimize both imputation and downstream tasks. This approach, as demonstrated to be effective in recent research (Qu et al., 2024), mitigates the complexities associated with multiple objectives, resulting in stable performance improvements, particularly in downstream tasks. The main contributions of this work are as follows.

- We present PIG, an MTL model that integrates imputation and prediction (classification or regression) for mixed-type tabular data, leveraging their interdependencies to enhance both tasks.

- We propose a GLL which identifies feature relationships, and a GRIL, which utilizes these relationships to optimize both imputation and downstream tasks.

- We propose a training strategy to optimize PIG for both imputation and prediction tasks, which begins with the pre-training of a GLL and a PL, followed by fine-tuning the entire model.

Through extensive experiments across diverse datasets that encompass both regression and classification downstream tasks, we rigorously evaluated the effectiveness of the proposed method. Several ablation studies further explains the significant impact of individual components within our model, thereby showing comprehensive and deliberate design of the proposed method.

## 2 RELATED WORK

MDI methods have evolved significantly over time, transitioning from simple statistical methods to more advanced machine learning and deep learning methods.

**Statistical Methods.** Early MDI methods included simple statistical methods such as mean, median, and mode imputation. While these methods are easy to implement, they often overlook relationships between features, leading to poor performance.

**Machine Learning Methods.** More sophisticated methods have emerged, such as multiple imputation by chained equations (MICE) (Van Buuren & Groothuis-Oudshoorn, 2011), $k$-nearest neighbors (KNN) (Troyanskaya et al., 2001), and matrix completion methods. MICE iteratively predicts missing values using estimators like Bayesian ridge or random forests, while KNN imputes missing values based on similar data points. Matrix completion methods, commonly employed in recommendation systems (Koren et al., 2009), have demonstrated strong performance in MDI by identifying and leveraging latent structures among samples and features. However, both MICE and KNN can face scalability issues and may struggle with large, complex datasets, and matrix completion methods are generally limited to numeric data, posing challenges with categorical variables.

**Deep Learning Methods.** Recently, deep learning methods have emerged as powerful alternatives for MDI. AEs (Gondara & Wang, 2018) and VAEs (Nazabal et al., 2020) project incomplete data into a latent space to reconstruct missing values. GAN-based methods, such as generative adversarial imputation networks (GAIN) (Yoon et al., 2018), utilize adversarial learning to generate plausible imputed data, reporting strong performance across various settings. Building on

Table 1: Summary of previous studies.

| Method | Datasets | Problem addressed | Missing rates (setting) |
|---|---|---|---|
| VAE (Nazabal et al., 2020) | 6 UCI datasets | Generative model, mixed-type data handling | $10\% \sim 50\%$ |
| GAIN (Yoon et al., 2018) | 5 UCI datasets | Accurate generative model | 20% (ablation: $10\% \sim 80\%$) |
| GRAPE (You et al., 2020) | 9 UCI datasets | Fully exploit information | 30% (ablation: $10\% \sim 70\%$) |
| MIRACLE (Kyono et al., 2021) | 10 UCI datasets | Causality preservation | 30% |
| HyperImpute (Jarrett et al., 2022) | 12 UCI datasets | Automatic model selection | $10\% \sim 70\%$ |

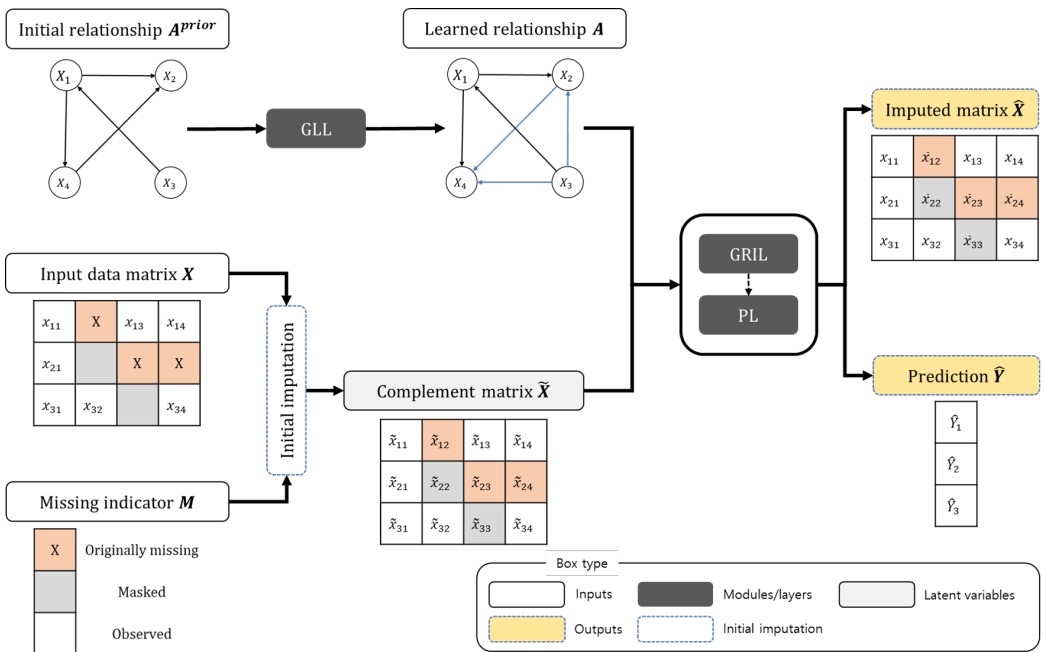

Figure 1: **Overall process of PIG.** The input data matrix $\mathbf{X}$, with missing entries indicated by the missing indicator $\mathbf{M}$, undergoes initial imputation to generate the complement matrix $\tilde{\mathbf{X}}$. The Graph Learning Layer (GLL) refines an initial feature relationship $\mathbf{A}^{prior}$ into a learned relationship matrix $\mathbf{A}$. The Graph Residual Imputation Layer (GRIL) leverages $\mathbf{A}$ and $\tilde{\mathbf{X}}$ to produce the imputed matrix $\hat{\mathbf{X}}$. Finally, the Prediction Layer (PL) uses $\hat{\mathbf{X}}$ for downstream tasks, yielding predictions $\hat{Y}$.

these advancements, GRAPE (You et al., 2020) leverages MTL to simultaneously tackle imputation and downstream tasks. Also, by applying GNNs to the tabular data represented as a bipartite graph, GRAPE implicitly captures meaningful feature relationships, enhancing imputation performance and facilitates improved outcomes in downstream tasks. MIRACLE (Kyono et al., 2021) is proposed to preserve the causal structure of data for more accurate imputation. It refines initial imputation from traditional methods by using neural networks while maintaining data dependencies. However, due to the specific purpose for data dependencies, MIRACLE suffers from performance degradation in inappropriate scenarios. At last, HyperImpute (Jarrett et al., 2022), a generalized iterative imputation framework, combines iterative imputation and recent powerful models; for each feature, it adaptively and automatically selects proper parameters and models, ranging from simple ML-based models to boosting trees and neural networks.

## 3 METHODS

### 3.1 OVERVIEW

This section briefly introduces the overall process of PIG from the original data matrix with missing values to the imputed data matrix and predicted values (Figure 1). Suppose an incomplete input data matrix $\boldsymbol{X} \in \mathbb{R}^{N \times d}$ and a missing indicator matrix $\boldsymbol{M} \in \{0, 1\}^{N \times d}$ where $N$ is the number of

observations and $d$ is the number of features. Each element $M_{ij}$ equals 1 if $X_{ij}$ is observed, and 0 if it is missing, for $i = 1, \ldots, N$ and $j = 1, \ldots, d$.

Initially, missing values in $\boldsymbol{X}$ should be filled using any simple imputation method. This initial imputation usually employs statistical and ML-based methods, yielding the complement matrix $\tilde{\boldsymbol{X}} \in \mathbb{R}^{N \times d}$ that contains no missing values. However, the imputed values can be further improved by utilizing informative feature relationships and considering both imputation and downstream task together.

Specifically, PIG uses residual learning (He et al., 2016) and feature relationships via graph convolution (Kipf & Welling, 2016) to correct $\tilde{\boldsymbol{X}}$. Then, PIG outputs the final imputation matrix $\hat{\boldsymbol{X}} \in \mathbb{R}^{N \times d}$, which is derived from the combination of observed values in $\boldsymbol{X}$ and imputed values for missing ones.

This final imputation matrix $\hat{\boldsymbol{X}}$ is fed to PL for downstream tasks. For simplicity, assume one downstream task, either a classification or regression. Then we can denote the target feature as $\boldsymbol{Y} \in \mathbb{R}^{N \times c}$ and its predicted values as $\hat{\boldsymbol{Y}} \in \mathbb{R}^{N \times c}$, where $c$ is the number of classes in the classification task and 1 in the regression task.

## 3.2 THE DETAILED MODEL ARCHITECTURE

PIG has three main layers (Figure 1): GLL, GRIL, and PL. The GLL learns the feature relationships, represented as a matrix $\boldsymbol{A} \in \mathbb{R}^{d \times d}$, that help both imputation and prediction. The GRIL takes $\tilde{\boldsymbol{X}}$ and $\boldsymbol{A}$ to correct the imputation residuals, then returns $\hat{\boldsymbol{X}}$ by filling in missing values of $\boldsymbol{X}$. Then, PL takes $\hat{\boldsymbol{X}}$ and predicts targets $\boldsymbol{Y}$ in downstream tasks. Each main layer will be described in the following subsections.

### 3.2.1 GLL: GRAPH LEARNING LAYER

Motivated by graph construction in various studies (Shi et al., 2019; Li et al., 2017; Wu et al., 2020; Yan et al., 2018), we devised GLL to adaptively infer feature relationships, which are typically unknown in real-world datasets. GLL utilizes independent representations: the destination-embedding matrix $\boldsymbol{E}_1 \in \mathbb{R}^{d \times d_E}$ and the source-embedding matrix $\boldsymbol{E}_2 \in \mathbb{R}^{d \times d_E}$ where $d_E$ is the dimensionality of feature embedding.

First, each embedding matrix is projected using $\boldsymbol{P}_1 \in \mathbb{R}^{d_E \times d_E}$ and $\boldsymbol{P}_2 \in \mathbb{R}^{d_E \times d_E}$ and activated with a scaled hyperbolic tangent function as $\boldsymbol{Z}_1 = \alpha \; tanh \left( \boldsymbol{E}_1 \boldsymbol{P}_1 \right)$ and $\boldsymbol{Z}_2 = \alpha \; tanh \left( \boldsymbol{E}_2 \boldsymbol{P}_2 \right)$ where $\alpha$ is a hyperparameter that controls sensitivity. Then, the feature relationship matrix $\boldsymbol{A}$ is computed by applying the sigmoid function to the inner product of the projected embeddings:

$$\boldsymbol{A} = sigmoid \left( \boldsymbol{Z}_1 \boldsymbol{Z}_2^T \right), \tag{1}$$

where each $A_{ij}$ indicates the informativeness of feature $j$ for imputing feature $i$. This relationship matrix $\boldsymbol{A}$ enables PIG to exploit feature dependencies for accurate imputation and prediction.

### 3.2.2 GRIL: GRAPH RESIDUAL IMPUTATION LAYER

GRIL takes the relationship matrix $\boldsymbol{A}$ and the complement matrix $\tilde{\boldsymbol{X}}$ as inputs and returns the final imputation matrix $\hat{\boldsymbol{X}}$. Initially, GRIL normalizes $\boldsymbol{A}$ as $\tilde{\boldsymbol{A}} = \boldsymbol{D}^{-1} \boldsymbol{A}$, where $\boldsymbol{D}$ is a diagonal matrix such that $D_{ii} = \sum_{j=1}^{d} A_{ij}$, $i \in \{1, \ldots, d\}$, and then uses $\tilde{\boldsymbol{A}}$ to aggregate relevant features for every feature. This aggregation step, called relationship graph convolution (RGC), is

$$\boldsymbol{H} = \tilde{\boldsymbol{X}} \tilde{\boldsymbol{A}} \boldsymbol{W}_1 + \boldsymbol{b}_1, \tag{2}$$

where $\boldsymbol{W}_1 \in \mathbb{R}^{d \times d}$ and $\boldsymbol{b}_1 \in \mathbb{R}^d$ are trainable parameters. Unlike usual graph convolution (Veličković et al., 2017; Hamilton et al., 2017), parameters of the features are directly aggregated. Then, residual connection (He et al., 2016) adds $\tilde{\boldsymbol{X}}$ to $\boldsymbol{H} \in \mathbb{R}^{N \times d}$ as $\bar{\boldsymbol{X}} = \sigma \left( \tilde{\boldsymbol{X}} + \boldsymbol{H} \right)$ where $\sigma$ can be *tanh* for numeric features and *sigmoid* for categorical features. With the use of residual learning at this layer, $\boldsymbol{H}$ serves as corrections that addresses the errors remaining in $\tilde{\boldsymbol{X}}$.

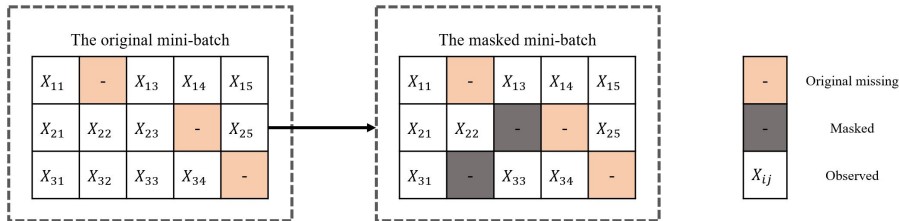

Figure 2: An example of random masking strategy. Observed values (white boxes) in the original mini-batch are randomly masked with probability $p_{mask}$.

At last, GRIL calculates the final imputation matrix $\hat{X}$ by maintaining observed values in $X$ and replacing missing values in $X$ with imputed values in $\bar{X}$, as

$$\hat{X} = X \odot M + \bar{X} \odot (1 - M),\qquad(3)$$

where $\odot$ is an element-wise multiplication.

### 3.2.3 PL: PREDICTION LAYER

After GRIL, PL uses the final imputation matrix $\hat{X}$ to conduct downstream tasks. Depending on the downstream task, the prediction is $\hat{Y} = softmax(\hat{X}W_2 + b_2)$ if the downstream task is classification, and $\hat{Y} = \hat{X}W_2 + b_2$ if it is regression, where $W_2 \in \mathbb{R}^{d \times c}$ and $b_2 \in \mathbb{R}^c$ are trainable parameters. Here, PL uses a one-layer fully-connected network (FCN) for the prediction, but the structure can be extended to increase complexity.

### 3.2.4 OBJECTIVE FUNCTION

The core idea of PIG is to use appropriate feature relationships to improve the solution for both imputation and prediction. The objective function is constructed over the three main layers (Figure 8 in Appendix E).

First, GLL computes the regularization loss $\mathcal{L}_{reg}$ to ensure that the inferred relationship matrix $A$ align with prior knowledge $A^{prior}$ about feature relationships. Specifically, it uses binary cross-entropy to assess how much $A$ deviates from $A^{prior}$. This loss encourages the model to deviate from $A^{prior}$ if deviations significantly enhance both imputation and prediction.

Second, GRIL computes the two types of losses: reconstruction loss $\mathcal{L}_{rec}$ and imputation loss $\mathcal{L}_{imp}$. The reconstruction loss ensures that the model accurately restores the actual values of the observed ones, whereas the imputation loss ensures that the model accurately guess true values of the missing ones. The imputation loss cannot be calculated directly because true values for unobserved ones are unknown. To solve this problem, observed values are randomly masked with probability $p_{mask}$ to simulate missing values. Then, the imputation loss is computed using these artificially masked values and the reconstruction loss is computed using only observed and unmasked values (Figure 2).

Specifically, GRIL distinguishes between numeric and categorical features by applying proper losses: squared differences for numeric features and cross-entropy loss for categorical features. This approach enables PIG to flexibly and accurately process mixed-type datasets.

Lastly, PL computes the downstream task loss $\mathcal{L}_{dt}$. Depending on whether the task is classification or regression, the last loss is calculated differently. It uses cross-entropy loss between the predicted labels $\hat{Y}$ and the true labels $Y$ for classification task, whereas it utilizes mean squared error (MSE) between the predicted values $\hat{Y}$ and the actual values $Y$ for regression task.

Finally, the total loss $\mathcal{L}$ of PIG is

$$\mathcal{L} = \lambda\mathcal{L}_{reg} + \mathcal{L}_{rec} + \mathcal{L}_{imp} + \mathcal{L}_{dt},\qquad(4)$$

where $\lambda$ is a term that adjusts the regularization penalty.

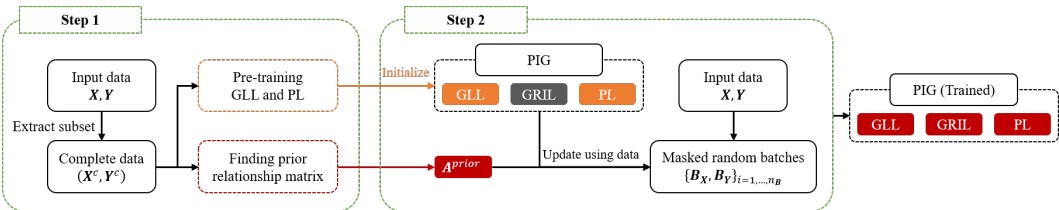

Figure 3: Two-step training of PIG. Orange boxes represent pre-training and red boxes represent the completion of training.

### 3.3 HEURISTIC ALGORITHM TO FIND PRIOR FEATURE RELATIONSHIPS

The feature relationships are unknown in many real-world data, so we propose a heuristic algorithm to estimate the prior feature relationships $A^{prior}$ (Algorithm 1). Imputation can be considered as a supervised learning task in which each feature is predicted by the others. If the $i$-th feature $X_{\cdot i}$ helps to predict another feature $X_{\cdot j}$, then a reasonable assumption is that $X_{\cdot i}$ is relevant in imputing $X_{\cdot j}$.

This relevance can be decently inferred using the feature importance of the decision tree algorithm (Hastie et al., 2009). Thus, we propose to use $d$ decision trees, each of which is fitted to predict each feature in data $X$. In this process, only complete data instances that have no missing values are used. Specifically, each decision tree is trained to predict $X_{\cdot i}^c$ using $X_{\cdot -i}^c$, which is obtained by excluding the $i$-th feature $X_{\cdot i}^c$. Here, the superscript $c$ indicates the subset of $X$ in which all features are observed.

In this case, the feature importance of $j$-th feature for predicting $i$-th feature, denoted as $FI_{ij}$, is the mean decrease in impurity of $X_{\cdot i}$. After the feature importance of every pair is computed, $A^{prior}$ is constructed as

$$A_{ij}^{prior} = \begin{cases} 1, \text{ if } FI_{ij} \geq \text{Average}(\boldsymbol{FI}_{i\cdot}) \\ 0, \text{ otherwise} \end{cases}, \tag{5}$$

where $\text{Average}(\boldsymbol{FI}_{i\cdot}) = \frac{1}{d-1} \sum_{j \neq i} FI_{ij}$. All diagonal elements $A_{ii}$ are set to 0.

These prior relationships may be a strong assumption on data, so $A$ will be updated during the training process only if a deviation from the prior is advantageous for reconstruction, imputation, or downstream tasks. The detailed steps of the algorithm are provided in Appendix D.

### 3.4 TRAINING STRATEGY

If parameter values were inappropriately initialized, then gradients of tightly related losses would have different directions. We carefully designed the two-step optimization strategy for PIG (Figure 3) to avoid this 'conflicting gradients' problem via pre-training. In step 1, the strategy pre-trains GLL and PL separately, and finds prior feature relationships $A^{prior}$. In this process, only complete data instances are used, so the pre-trained parameters may be sub-optimal. In step 2, GLL and PL are initialized with the pre-trained parameters and then all the parameters in PIG are jointly optimized using the total loss $\mathcal{L}$. This strategy succesfully guides PIG to an improved solution (see Table 3).

## 4 EXPERIMENTS

### 4.1 DATASETS AND EXPERIMENTAL SETTINGS

To evaluate the performance of PIG, we conducted various experiments using nine benchmark datasets from the UCI Machine Learning Repository (Lichman, 2013), as in previous studies on MDI for tabular datasets (Table 1); Among them, three datasets involve the regression downstream task and six involve the classification downstream task (Table 4). In a few datasets, we removed data instances with missing values to make these datasets evaluable. We then randomly generated missing values in the datasets to simulate data that have missing values. Finally, we pre-processed all the numeric features to be scaled to $[-1, 1]$, and categorical features to be one-hot encoded.

We first evaluated imputation and prediction accuracy. For this experiment, we randomly removed 20% of elements and used them as originally-missing values, and then calculated the performance.

For the imputation task, we measured MSE for numeric features and CE for categorical features. For the regression downstream task, we assessed mean absolute error (MAE) and MSE, while for the classification downstream task, we assessed accuracy, precision, recall, and F1 score.

We conducted an ablation study to test the effect of GLL, the initial imputation strategy for $\tilde{X}$, and the pre-training strategy. At last, We also conducted the sensitivity analysis to test how PIG is sensitive to missing rates, adjusting the ratio from 5% to 50% (see Appendix K).

We repeated every experiment 10 times and reported the average accuracy on the test set. For each repetition, We first randomly divided the datasets into 70% for training, 20% for validation, and 10% for testing. Then, we introduced random missing values into the training dataset.

We set hyperparameters of PIG as follows. The initial imputation method is Soft-Impute (SI), as described by Mazumder et al. (2010), unless otherwise stated. We set the dimensionality $d_E$ of feature embeddings to $\lfloor d/2 \rfloor$ differently according to datasets, where $\lfloor \cdot \rfloor$ is the floor function. The scaling parameter $\alpha$ that controls the sensitivity in GLL was set to 5 and the regularization parameter $\lambda$ (Equation 4) that controls the penalty of the regularization loss to 0.01. We trained PIG for 60 epochs using Adam optimizer (Kingma & Ba, 2014) with learning rate 0.01, and applied early stopping of 10-epoch-patience. We set a random masking probability $p_{mask}$ to 20% for all datasets.

We compared PIG with some existing imputation methods and their MTL versions (if extendable) to see whether the improvement of our method is attributed to only MTL. MTL versions of existing models are implemented by modifying each method to learn imputation and prediction simultaneously as in PIG. In addition, a model without the MTL structure first learns imputation with a complete matrix and then trains PL by using imputed data, where the prediction layer is one-layer FCN as in PIG. As baseline models, we used SI (Mazumder et al., 2010), AE, VAE, GRAPE (You et al., 2020), MIRACLE (Kyono et al., 2021), and HyperImpute (Jarrett et al., 2022). A brief description of these baseline models is in Appendix C.

## 4.2 IMPUTATION AND PREDICTION ACCURACY

We conducted experiments on nine benchmark datasets and measured the accuracies of the imputation and downstream tasks. Three datasets (i.e., "Energy", "Abalone", and "Diabetes") have a regression downstream task and the rest (i.e., "Spam", "Mobile", "Pulsar", "Breast Cancer", "Faults", and "Wine") have a classification downstream task.

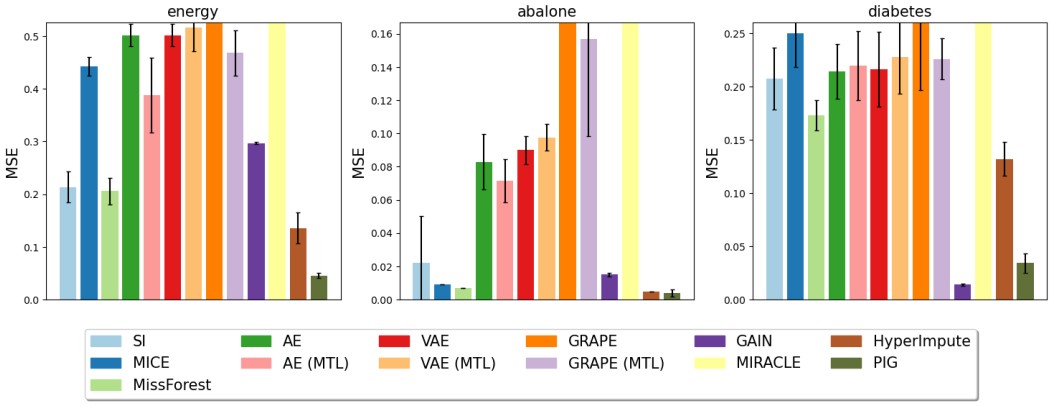

Figure 4: The imputation performance across three regression datasets. Each chart displays the results for a specific dataset. To reduce complexity, only numerical imputations are visualized.

PIG showed the best imputation and regression accuracy for three regression datasets (Figure 4 and 5). PIG consistently showed best performance, except for a second-place on the "Diabetes" dataset. In this dataset, GAIN outperformed our methods in both imputation and regression performance;

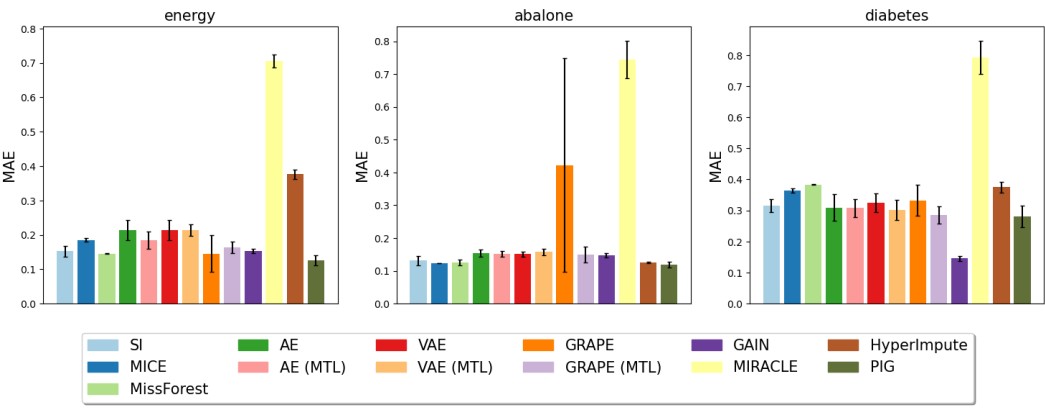

Figure 5: The regression performance for three datasets. Each chart displays the results for a specific dataset.

however, it was unstable in that its imputation performance was worse than that of traditional methods like SI and MissForest in other datasets. All experimental results are detailed in Appendix I.

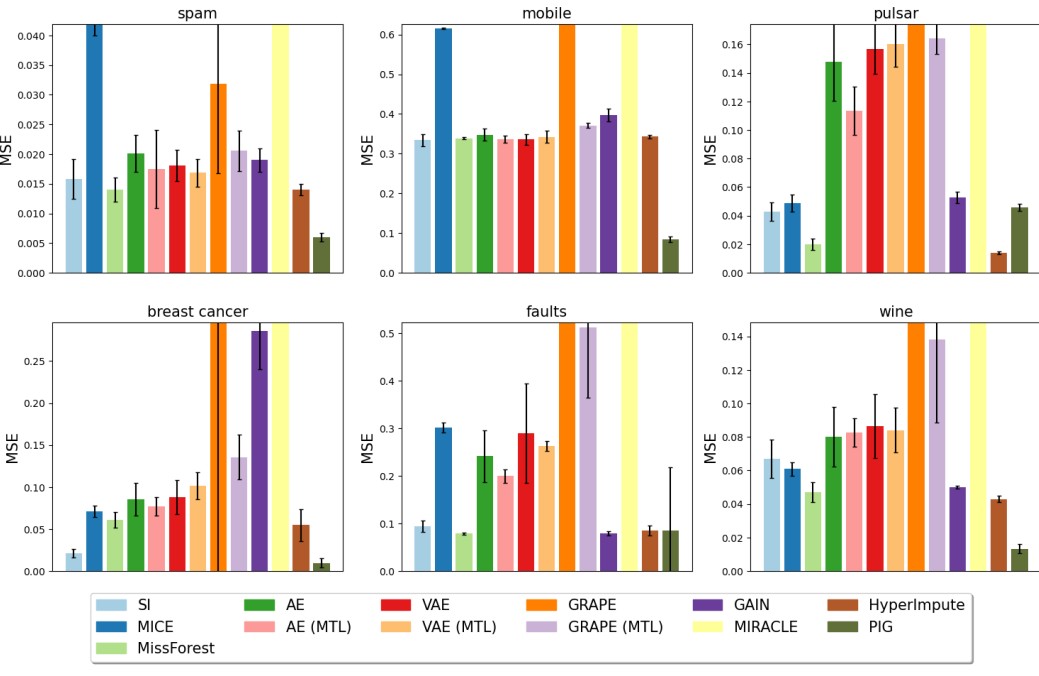

Figure 6: The imputation performance for six classification datasets. Each chart displays the results for a specific dataset. To reduce complexity, only numerical imputations are visualized.

In addition, PIG showed superior performance for the six classification datasets (Figure 6 and 7). PIG achieved first place in all classification datasets, with the exception of the "Wine" dataset, where PIG ranked second in terms of classification performance despite the best imputation performance. All experimental results are detailed in Appendix J.

In both imputation and prediction tasks, regardless of regression or classification, PIG demonstrated superior performance while achieving the top average ranking among all baseline models (Table 5 in Appendix F). Especially, the gap with the runner-up method is fairly large in prediction tasks compared to imputation tasks. Additionally, methods with MTL demonstrated similar or slightly degraded imputation performance compared to other methods, but exhibited superior predictive performance in both regression and classification. This implies that considering downstream tasks is

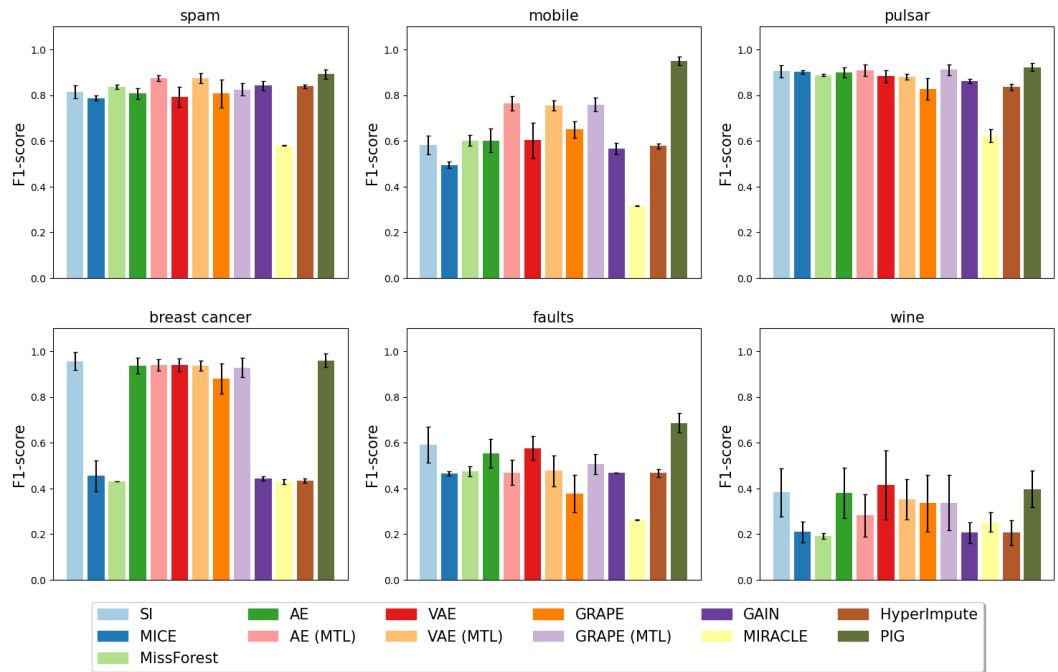

Figure 7: The classification performance for six datasets. Each chart displays the results for a specific dataset.

crucial when imputation is necessary due to missing data. Among the MTL methods, PIG outperformed the other methods, which means that considering feature relationships is crucial for addressing tabular data with missing values. In both regression and classification tasks, PIG demonstrated significantly improved imputation performance compared to SI, which is used to generate initial imputation matrix in PIG. This means that PIG greatly contributed to the enhancement of imputation performance by utilizing graph convolution with adaptive feature relationships.

### 4.3 ABLATION STUDY

We conducted a few ablation studies to evaluate the individual effect of the GLL, initial imputation strategies, and pre-training strategies. We explored their effects on both types of downstream tasks, regression and classification, using "Diabetes" and "Mobile" datasets, respectively. Due to space constraints, the ablation study on initial imputation strategies is provided in Appendix H, and the experimental settings beyond the focus of each ablation study are detailed in Section 4.2.

#### 4.3.1 THE EFFECT OF GLL

We compared PIG with and without GLL and showed that the feature relationships changed after training PIG ended. We visualized the difference in feature relationships between $A$ and $A^{prior}$ (Figure 9 in Appendix G). The prevalence of consistent relationships (black arrows, Figure 9) highlights the stability of our heuristic approach (Section 3.3). Furthermore, the presence of red and blue arrows implies that GLL facilitates adaptive learning of feature relationships. These changes also led to improvement of performance in both imputation and downstream tasks (Table 2), demonstrating that GLL contributed the improvement of imputation and downstream tasks by adjusting feature relationships in a data-driven manner.

#### 4.3.2 THE EFFECT OF PRE-TRAINING STRATEGY

Finally, we assessed the impact of our pre-training strategy by comparing the performance of imputation and prediction tasks with and without pre-training (Table 3). It appears that the pre-training strategy is not as effective for imputation tasks, as PIG with pre-training showed more accurate per-

Table 2: Ablation results to measure the impact of GLL for two datasets: Mobile (classification) and Diabetes (regression).

| Dataset | Models | Imputation Task | | Regression Task | |
|---|---|---|---|---|---|
| | | MSE | | MSE | MAE |
| Diabetes | PIG | **0.034** $\pm$ **0.009** | | **0.122** $\pm$ **0.027** | **0.280** $\pm$ **0.035** |
| | PIG w/o GLL | 0.041 $\pm$ 0.006 | | 0.125 $\pm$ 0.024 | 0.288 $\pm$ 0.037 |

| Dataset | Models | Imputation Task | | Classification Task | | | |
|---|---|---|---|---|---|---|---|
| | | MSE | CE | Accuracy | Recall | Precision | F1-score |
| Mobile | PIG | **0.084** $\pm$ **0.007** | **0.222** $\pm$ **0.042** | **0.952** $\pm$ **0.017** | **0.951** $\pm$ **0.018** | **0.952** $\pm$ **0.017** | **0.950** $\pm$ **0.018** |
| | PIG w/o GLL | 0.101 $\pm$ 0.005 | 0.344 $\pm$ 0.016 | 0.938 $\pm$ 0.014 | 0.940 $\pm$ 0.014 | 0.938 $\pm$ 0.014 | 0.938 $\pm$ 0.014 |

Table 3: Ablation results to measure the impact of pre-training.

| Data | Models | Imputation Task | | Regression Task | |
|---|---|---|---|---|---|
| | | MSE | CE | MSE | MAE |
| Energy | PIG | 0.046 $\pm$ 0.005 | - | **0.031** $\pm$ **0.006** | **0.126** $\pm$ **0.015** |
| | PIG w/o pre-train | **0.043** $\pm$ **0.007** | | 0.080 $\pm$ 0.065 | 0.194 $\pm$ 0.084 |
| Abalone | PIG | **0.004** $\pm$ **0.002** | **0.100** $\pm$ **0.013** | **0.027** $\pm$ **0.005** | **0.118** $\pm$ **0.009** |
| | PIG w/o pre-train | 0.007 $\pm$ 0.007 | 0.124 $\pm$ 0.013 | 0.029 $\pm$ 0.007 | 0.121 $\pm$ 0.009 |
| Diabetes | PIG | 0.034 $\pm$ 0.009 | - | **0.122** $\pm$ **0.027** | **0.280** $\pm$ **0.035** |
| | PIG w/o pre-train | **0.028** $\pm$ **0.006** | | 0.136 $\pm$ 0.026 | 0.299 $\pm$ 0.037 |

| Data | Models | Imputation Task | | Classification Task | | | |
|---|---|---|---|---|---|---|---|
| | | MSE | CE | Accuracy | Recall | Precision | F1-score |
| Spam | PIG | 0.006 $\pm$ 0.001 | - | **0.899** $\pm$ **0.020** | **0.823** $\pm$ **0.022** | **0.901** $\pm$ **0.020** | **0.892** $\pm$ **0.022** |
| | PIG w/o pre-train | **0.005** $\pm$ **0.001** | | 0.849 $\pm$ 0.017 | **0.823** $\pm$ **0.022** | 0.859 $\pm$ 0.017 | 0.828 $\pm$ 0.021 |
| Mobile | PIG | **0.084** $\pm$ **0.007** | **0.222** $\pm$ **0.042** | **0.952** $\pm$ **0.017** | **0.951** $\pm$ **0.018** | **0.952** $\pm$ **0.017** | **0.950** $\pm$ **0.018** |
| | PIG w/o pre-train | 0.061 $\pm$ 0.006 | 0.393 $\pm$ 0.011 | 0.712 $\pm$ 0.034 | 0.719 $\pm$ 0.034 | 0.712 $\pm$ 0.037 | 0.691 $\pm$ 0.036 |
| Pulsar | PIG | 0.046 $\pm$ 0.002 | - | **0.923** $\pm$ **0.018** | **0.923** $\pm$ **0.018** | **0.926** $\pm$ **0.017** | **0.922** $\pm$ **0.018** |
| | PIG w/o pre-train | **0.014** $\pm$ **0.002** | | 0.909 $\pm$ 0.027 | 0.909 $\pm$ 0.025 | 0.913 $\pm$ 0.029 | 0.906 $\pm$ 0.029 |
| Breast Cancer | PIG | **0.010** $\pm$ **0.006** | - | **0.962** $\pm$ **0.029** | **0.956** $\pm$ **0.033** | **0.966** $\pm$ **0.025** | **0.960** $\pm$ **0.030** |
| | PIG w/o pre-train | 0.020 $\pm$ 0.006 | | 0.714 $\pm$ 0.138 | 0.536 $\pm$ 0.157 | 0.567 $\pm$ 0.264 | 0.473 $\pm$ 0.182 |
| Faults | PIG | **0.085** $\pm$ **0.134** | - | 0.692 $\pm$ 0.026 | **0.714** $\pm$ **0.048** | 0.690 $\pm$ 0.042 | **0.686** $\pm$ **0.042** |
| | PIG w/o pre-train | 0.153 $\pm$ 0.219 | | **0.693** $\pm$ **0.042** | 0.705 $\pm$ 0.055 | **0.755** $\pm$ **0.051** | 0.668 $\pm$ 0.064 |
| Wine | PIG | 0.013 $\pm$ 0.003 | - | 0.561 $\pm$ 0.020 | 0.405 $\pm$ 0.078 | 0.775 $\pm$ 0.037 | 0.397 $\pm$ 0.080 |
| | PIG w/o pre-train | **0.009** $\pm$ **0.002** | | **0.569** $\pm$ **0.056** | **0.546** $\pm$ **0.059** | **0.810** $\pm$ **0.035** | **0.517** $\pm$ **0.068** |

formance in only four out of nine datasets. However, the pre-training strategy significantly enhanced regression or classification performance in almost all datasets, with the exception of the "Wine" dataset. This implies that pre-training effectively stabilizes the two different objectives, imputation and prediction, by properly setting the initial parameters in GLL and PL.

## 5 CONCLUSION

This paper has presented PIG, a framework for MTL of imputation and prediction as a downstream task. We also proposed a heuristic method to easily find feature relationships that may be helpful to imputation and downstream tasks. The framework can exploit these relationships in graph convolution to correct the initial imputation, then updates the relationships if necessary to improve the accuracy of both tasks. To achieve the mentioned purposes, the objective function of PIG has several loss terms, which are difficult to optimize simultaneously; thus, we suggested first pre-training GLL and PL and then jointly optimizing the whole PIG. This training strategy successfully balances several tasks. Empirically, PIG achieved the best performance in imputation and prediction tasks across nine datasets, leading to the average rank of 1.78 for imputation task and 1.22 for downstream tasks. Consequently, PIG can be the robust and versatile approach in handling tabular data with missing values with specific downstream tasks.

## 6 REPRODUCIBILITY

The hyperparameters, data preprocessing steps, and experiment settings are provided in Section 4.1 of the main paper, while the details of the datasets are summarized in Table 4 in the appendix. For reproducibility, we will make the source code publicly available.

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

## A  TYPES OF MISSINGNESS

Missing data are of three types: Missing At Random (MAR), Missing Not At Random (MNAR), and Missing Completely At Random (MCAR). MAR refers to the case that missingness depends on the observed features, MNAR refers to the case that missingness depends on both observed and unobserved data. In MCAR case, missingness is independent of both observed and unobserved data. For example, consider two observed input features $x_1$ and $x_2$, and an unobserved feature $z$. If the missingness of $x_1$ depends only on $x_2$ or vice versa, the data is MAR. If the missingness of $x_1$ or $x_2$ additionally depends on $z$, the data is MNAR. Finally, if missingness is independent of other features, it is MCAR.

Among the three types, MCAR represents the simplest type of missing data and thus enables the evaluation of imputation models without the introduction of biases unlike MAR and MNAR. Due to this simplicity, most existing studies on MDI are based on the MCAR assumption as a stepping stone towards more complex scenarios of MAR and MNAR. Therefore, this study also focuses on the MCAR case to make our results be directly comparable with those studies.

## B  DETAILS OF DATASETS.

Table 4: Details of datasets.

| Task | Dataset | # instances | # features | |
|------|---------|-------------|------------|------|
| | | | Numeric | Categorical |
| Regression | Energy | 1,296 | 28 | 0 |
| | Abalone | 4,177 | 7 | 1 |
| | Diabetes | 442 | 10 | 0 |
| Classification | Spam | 4,601 | 57 | 0 |
| | Mobile | 2,000 | 14 | 6 |
| | Pulsar | 12,528 | 9 | 0 |
| | Breast Cancer | 569 | 30 | 0 |
| | Faults | 1,941 | 33 | 0 |
| | Wine | 1,143 | 12 | 0 |

## C  BASELINE MODEL DESCRIPTION

- Soft-Impute (SI) (Mazumder et al., 2010): a matrix completion method that uses the nuclear norm as a regularization loss.

- Auto-Encoder (AE): A traditional DL method that focuses on the reconstruction loss to impute missing values.

- Variational Auto-Encoder (VAE): A deep generative method that has a similar model structure to AE but trains the model weights by maximizing the evidence lower bound

- GRAPE (You et al., 2020): A GNN-based representation learning method to impute missing values. This method constructs a bipartite graph of $N$ data instance nodes and $d$ feature nodes. This method can be itself an MTL approach.

- MIRACLE (Kyono et al., 2021): A neural network-based imputation method that refines imputation results by incorporating data dependencies.

- HyperImpute (Jarrett et al., 2022): A generalized iterative imputation framework that adaptively selects the most suitable model, ranging from traditional to recent models, for each feature.

MTL versions of AE, VAE, and GRAPE are denoted as AE (MTL), VAE (MTL), and GRAPE (MTL), respectively. Both PIG and MTL versions of baseline models used a one-layer FCN for fair comparisons. AE and VAE require a complete matrix as an input, so SI first initializes an incomplete matrix $X$ for those two methods. Hyperparameters were set to as suggested by their authors (You et al., 2020) or determined experimentally.

## D  HEURISTIC ALGORITHM FOR PRIOR RELATIONSHIPS

---

**Algorithm 1** The proposed heuristic algorithm for prior relationships

---

**Inputs:**
$X^c$: a subset of $X$ that contains only data instances with no missing values.

1: Initialize prior feature relationship matrix $A^{prior} \in \mathbb{R}^{d \times d}$
2: **for** $i = 1, \ldots, d$ **do**
3:      $X^c_{\cdot i} \leftarrow i$-th column of $X^c$
4:      $X^c_{\cdot -i} \leftarrow$ a subset of $X^c$ derived by excluding $i$-th column $X^c_{\cdot i}$
5:      Fit a decision tree $DT_i$ that predicts $X^c_{\cdot i}$ from $X^c_{\cdot -i}$
6:      $FI_{i\cdot} \leftarrow$ feature importance derived from $DT_i$       $\triangleright FI_{ii}$ is set to zero.
7:      **for** $j = 1, \ldots, d$ **do**
8:          $A^{prior}_{ij} \leftarrow \begin{cases} 1, & \text{if } FI_{ij} \geq \text{Average}(FI_{i\cdot}) \\ 0, & \text{otherwise} \end{cases}$
9:      **end for**
10: **end for**

11: **return** $A^{prior}$

---

## E  OBJECTIVE FUNCTION DESIGN OF PIG

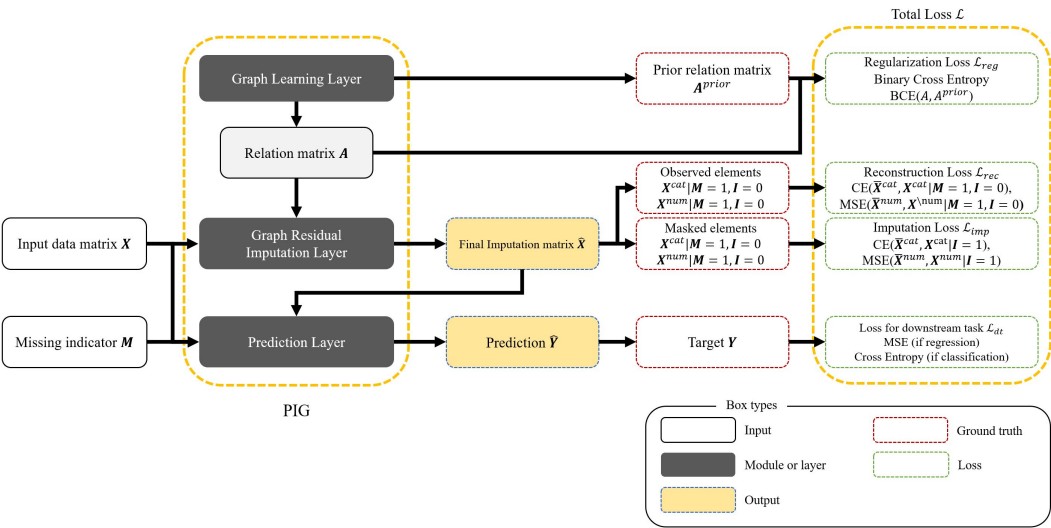

Figure 8: Objective function design of PIG.

# F    AVERAGE RANKINGS OF IMPUTATION AND PREDICTION

Table 5: Average rankings of imputation and prediction.

| Task | Method | Imputation | | Regression | | Classification | |
|------|--------|------------|------|------------|------|----------------|----------|
| | | MSE | CE | MSE | MAE | Accuracy | F1 score |
| **Regression** | SI | 5.00 | 7.00 | 5.00 | 5.33 | - | - |
| | MICE | 7.33 | 10.00 | 6.67 | 6.67 | - | - |
| | MissForest | 3.33 | 11.00 | 6.33 | 5.67 | - | - |
| | AE | 7.67 | 2.00 | 7.67 | 8.67 | - | - |
| | VAE | 8.33 | 4.00 | 7.67 | 8.67 | - | - |
| | AE (MTL) | 7.00 | 5.00 | 6.00 | 7.00 | - | - |
| | VAE (MTL) | 10.33 | 6.00 | 7.00 | 8.00 | - | - |
| | GRAPE (MTL) | 9.33 | 3.00 | 4.33 | 5.33 | - | - |
| | GRAPE | 12.00 | 8.00 | 11.00 | 7.67 | - | - |
| | MIRACLE | 13.00 | 13.00 | 13.00 | 13.00 | - | - |
| | HyperImpute | 2.33 | 12.00 | 8.00 | 9.00 | - | - |
| | GAIN | 3.67 | 9.00 | 5.33 | 4.00 | - | - |
| | PIG | **1.33** | **1.00** | **1.33** | **1.33** | - | - |
| **Classification** | SI | 3.67 | 3.00 | - | - | 5.17 | 4.67 |
| | MICE | 8.00 | 8.00 | - | - | 9.00 | 9.83 |
| | MissForest | 2.83 | 9.00 | - | - | 7.33 | 8.67 |
| | AE | 7.67 | 3.00 | - | - | 6.67 | 6.00 |
| | VAE | 7.67 | 2.00 | - | - | 5.33 | 5.33 |
| | AE (MTL) | 5.83 | 6.00 | - | - | 3.67 | 4.50 |
| | VAE (MTL) | 7.83 | 12.00 | - | - | 5.83 | 5.17 |
| | GRAPE (MTL) | 10.33 | 7.00 | - | - | 5.00 | 5.00 |
| | GRAPE | 11.83 | 12.00 | - | - | 10.00 | 8.83 |
| | MIRACLE | 13.00 | 5.00 | - | - | 13.00 | 12.33 |
| | HyperImpute | 2.83 | 10.00 | - | - | 9.00 | 9.67 |
| | GAIN | 6.67 | 11.00 | - | - | 8.33 | 9.17 |
| | PIG | **1.83** | **1.00** | - | - | **1.17** | **1.17** |

## G THE VISUALIZATION OF CHANGES IN FEATURE RELATIONSHIPS BY GLL

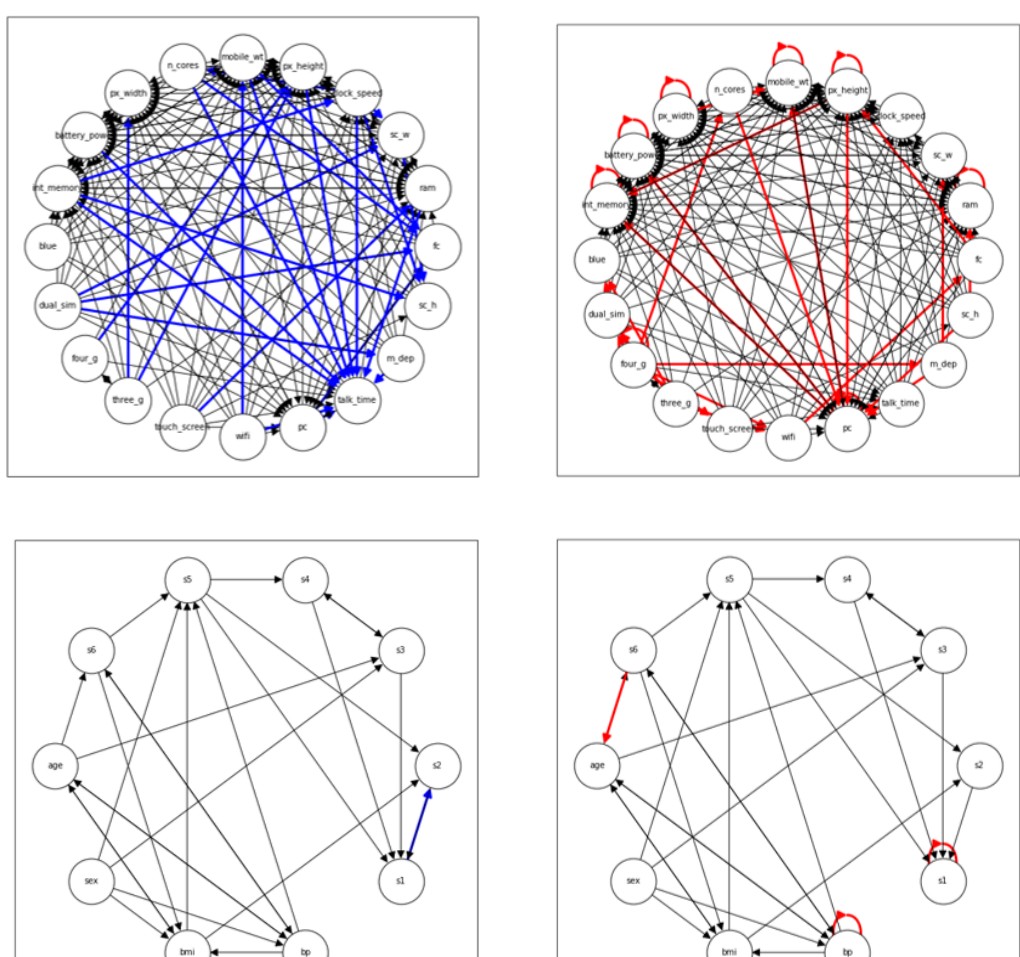

Figure 9: The visualization of changes in feature relationships by GLL (Upper section: "Mobile" dataset, lower section: "Diabetes" dataset). Circles represent features in each dataset, and arrows indicate the asymmetric relationships between these features, where the features at the tail of arrows significant affect the features at the head. Black arrows represent feature relationships that are consistent in both $A^{prior}$ and $A$. The blue arrows indicate feature relationships that were present in $A^{prior}$ but deleted in $A$, while red arrows represent feature relationships that were not in $A^{prior}$ but have been newly added to $A$. We created the feature relationship graph by thresholding $A$ at a value of 0.5.

# H ABLATION STUDY: THE EFFECT OF THE INITIAL IMPUTATION METHOD

We tested the performance of PIG using different initial imputation methods (Table 6). We compared four simple imputation strategies: SI, MEAN, MEDIAN, and CONSTANT. MEAN and MEDIAN use each feature's mean or median value to fill in the missing values, and CONSTANT replaces missing values with zeros. We first measured the imputation accuracy of these four imputation methods and tested downstream task accuracy using the data imputed by these methods (the prediction model is trained in a separate step). Then we trained four PIG models that used different initial imputation methods and measured their imputation and downstream task accuracy. Each model is denoted as PIG (imputation method), such as PIG (MEAN).

The results of Mobile dataset show that initial imputation methods can make meaningful differences in both classification performance and imputation accuracy of categorical features. In the results of Diabetes dataset, imputation accuracy of numerical features varied according to initial imputation methods. However, PIG increased the performance of both imputation and downstream tasks regardless of initial imputation methods, compared to two-phase training methods that use four simple imputation strategies.

We also compared PIG with four simple imputation methods to other baseline models. PIG achieved accuracy much higher than the baseline models regardless of the initial imputation method. This means that PIG greatly contributed to the enhancement of imputation performance from the initial imputation matrix by utilizing graph convolution with adaptive feature relationship.

Table 6: Comparison of PIG models with different initial imputation methods for two datasets: Diabetes (regression) and Mobile (classification).

| Data | Models | Imputation Task | Regression Task | |
|------|--------|-----------------|-----------------|---|
| | | MSE | MSE | MAE |
| Diabetes | PIG (CONSTANT) | $0.047 \pm 0.007$ | $0.127 \pm 0.017$ | $0.289 \pm 0.021$ |
| | PIG (MEAN) | $0.044 \pm 0.010$ | $0.130 \pm 0.021$ | $0.289 \pm 0.030$ |
| | PIG (MEDIAN) | $0.063 \pm 0.015$ | $\mathbf{0.119 \pm 0.016}$ | $0.281 \pm 0.027$ |
| | PIG (SI) | $\mathbf{0.034 \pm 0.009}$ | $0.122 \pm 0.027$ | $\mathbf{0.280 \pm 0.035}$ |
| | CONSTANT | $0.251 \pm 0.031$ | $0.156 \pm 0.024$ | $0.324 \pm 0.031$ |
| | MEAN | $0.230 \pm 0.050$ | $0.135 \pm 0.021$ | $0.302 \pm 0.030$ |
| | MEDIAN | $0.305 \pm 0.067$ | $\mathbf{0.119 \pm 0.022}$ | $\mathbf{0.280 \pm 0.027}$ |
| | SI | $0.207 \pm 0.029$ | $0.146 \pm 0.018$ | $0.315 \pm 0.020$ |
| | AE | $0.214 \pm 0.026$ | $0.144 \pm 0.033$ | $0.309 \pm 0.042$ |
| | VAE | $0.216 \pm 0.035$ | $0.155 \pm 0.026$ | $0.325 \pm 0.031$ |
| | AE (MTL) | $0.220 \pm 0.032$ | $0.140 \pm 0.027$ | $0.308 \pm 0.028$ |
| | VAE (MTL) | $0.227 \pm 0.034$ | $0.141 \pm 0.029$ | $0.302 \pm 0.033$ |
| | GRAPE (MTL) | $0.226 \pm 0.019$ | $0.123 \pm 0.022$ | $0.285 \pm 0.028$ |

| Data | Models | Imputation Task | | Classification Task | | | |
|------|--------|-----------------|---|---------------------|---|---|---|
| | | MSE | CE | Accuracy | Recall | Precision | F1-score |
| Mobile | PIG (CONSTANT) | $0.081 \pm 0.004$ | $0.278 \pm 0.009$ | $0.922 \pm 0.022$ | $0.924 \pm 0.021$ | $0.921 \pm 0.023$ | $0.921 \pm 0.022$ |
| | PIG (MEAN) | $\mathbf{0.078 \pm 0.004}$ | $\mathbf{0.202 \pm 0.023}$ | $0.922 \pm 0.019$ | $0.923 \pm 0.017$ | $0.920 \pm 0.023$ | $0.920 \pm 0.021$ |
| | PIG (MEDIAN) | $0.079 \pm 0.004$ | $0.319 \pm 0.021$ | $0.911 \pm 0.016$ | $0.912 \pm 0.020$ | $0.912 \pm 0.015$ | $0.909 \pm 0.018$ |
| | PIG (SI) | $0.084 \pm 0.007$ | $0.222 \pm 0.042$ | $\mathbf{0.952 \pm 0.017}$ | $\mathbf{0.951 \pm 0.018}$ | $\mathbf{0.952 \pm 0.017}$ | $\mathbf{0.950 \pm 0.018}$ |
| | CONSTANT | $0.382 \pm 0.014$ | $0.693 \pm 0.000$ | $0.598 \pm 0.031$ | $0.601 \pm 0.032$ | $0.594 \pm 0.033$ | $0.592 \pm 0.035$ |
| | MEAN | $0.346 \pm 0.014$ | $0.710 \pm 0.019$ | $0.594 \pm 0.038$ | $0.590 \pm 0.036$ | $0.591 \pm 0.034$ | $0.586 \pm 0.035$ |
| | MEDIAN | $0.346 \pm 0.011$ | $0.725 \pm 0.032$ | $0.599 \pm 0.065$ | $0.597 \pm 0.063$ | $0.601 \pm 0.064$ | $0.592 \pm 0.065$ |
| | SI | $0.334 \pm 0.015$ | $0.676 \pm 0.012$ | $0.590 \pm 0.041$ | $0.594 \pm 0.038$ | $0.588 \pm 0.040$ | $0.582 \pm 0.041$ |
| | AE | $0.348 \pm 0.015$ | $0.676 \pm 0.008$ | $0.609 \pm 0.047$ | $0.610 \pm 0.048$ | $0.605 \pm 0.053$ | $0.601 \pm 0.052$ |
| | VAE | $0.336 \pm 0.013$ | $0.665 \pm 0.009$ | $0.616 \pm 0.077$ | $0.611 \pm 0.070$ | $0.606 \pm 0.076$ | $0.603 \pm 0.077$ |
| | AE (MTL) | $0.336 \pm 0.009$ | $0.695 \pm 0.026$ | $0.758 \pm 0.036$ | $0.760 \pm 0.034$ | $0.791 \pm 0.030$ | $0.763 \pm 0.032$ |
| | VAE (MTL) | $0.343 \pm 0.015$ | $0.727 \pm 0.034$ | $0.750 \pm 0.021$ | $0.750 \pm 0.020$ | $0.777 \pm 0.022$ | $0.754 \pm 0.022$ |
| | GRAPE (MTL) | $0.371 \pm 0.006$ | $0.697 \pm 0.001$ | $0.757 \pm 0.033$ | $0.756 \pm 0.030$ | $0.785 \pm 0.034$ | $0.758 \pm 0.030$ |

## I EXPERIMENTAL RESULTS FOR THREE REGRESSION DATASETS

Table 7: The regression and imputation performance for three datasets. MAE stands for mean absolute error. Bold: best score. Underline: runner-up.

| Data | Models | Regression Task | | Imputation Task | |
|------|--------|-----------------|-----|-----------------|-----|
| | | MSE | MAE | MSE | CE |
| Energy | SI | $0.045 \pm 0.011$ | $0.152 \pm 0.016$ | $0.213 \pm 0.029$ | - |
| | MICE | $0.074 \pm 0.004$ | $0.185 \pm 0.005$ | $0.443 \pm 0.018$ | - |
| | MissForest | $\underline{0.041} \pm 0.002$ | $0.145 \pm 0.001$ | $0.206 \pm 0.025$ | - |
| | AE | $0.082 \pm 0.018$ | $0.214 \pm 0.029$ | $0.502 \pm 0.021$ | - |
| | VAE | $0.082 \pm 0.018$ | $0.214 \pm 0.029$ | $0.502 \pm 0.021$ | - |
| | GRAPE | $0.278 \pm 0.044$ | $0.145 \pm 0.053$ | $0.915 \pm 0.384$ | - |
| | GAIN | $0.044 \pm 0.011$ | $0.153 \pm 0.006$ | $0.297 \pm 0.002$ | - |
| | MIRACLE | $0.953 \pm 0.025$ | $0.706 \pm 0.019$ | $14.963 \pm 0.142$ | - |
| | HyperImpute | $0.205 \pm 0.015$ | $0.376 \pm 0.013$ | $\underline{0.136} \pm 0.029$ | - |
| | AE (MTL) | $0.062 \pm 0.017$ | $0.184 \pm 0.025$ | $0.388 \pm 0.071$ | - |
| | VAE (MTL) | $0.081 \pm 0.014$ | $0.213 \pm 0.016$ | $0.516 \pm 0.044$ | - |
| | GRAPE (MTL) | $0.042 \pm 0.006$ | $0.163 \pm 0.016$ | $0.468 \pm 0.043$ | - |
| | PIG | $\mathbf{0.031} \pm \mathbf{0.006}$ | $\mathbf{0.126} \pm \mathbf{0.015}$ | $\mathbf{0.046} \pm \mathbf{0.005}$ | - |
| Abalone | SI | $0.033 \pm 0.007$ | $0.132 \pm 0.015$ | $0.022 \pm 0.028$ | $0.699 \pm 0.010$ |
| | MICE | $0.033 \pm 0.001$ | $\underline{0.124} \pm 0.000$ | $0.009 \pm 0.0000$ | $0.825 \pm 0.026$ |
| | MissForest | $0.035 \pm 0.004$ | $0.125 \pm 0.008$ | $0.007 \pm 0.000$ | $0.831 \pm 0.002$ |
| | AE | $0.045 \pm 0.008$ | $0.154 \pm 0.011$ | $0.083 \pm 0.017$ | $\underline{0.606} \pm 0.033$ |
| | VAE | $0.040 \pm 0.003$ | $0.151 \pm 0.007$ | $0.090 \pm 0.009$ | $0.648 \pm 0.017$ |
| | GRAPE | $0.428 \pm 0.197$ | $0.422 \pm 0.326$ | $1.062 \pm 0.651$ | $0.807 \pm 0.102$ |
| | GAIN | $0.050 \pm 0.050$ | $0.147 \pm 0.007$ | $0.015 \pm 0.001$ | $0.824 \pm 0.001$ |
| | MIRACLE | $0.939 \pm 0.123$ | $0.745 \pm 0.057$ | $11.370 \pm 0.174$ | $1.085 \pm 0.011$ |
| | HyperImpute | $\underline{0.032} \pm 0.002$ | $0.126 \pm 0.002$ | $\underline{0.005} \pm 0.000$ | $0.836 \pm 0.000$ |
| | AE (MTL) | $0.045 \pm 0.005$ | $0.152 \pm 0.009$ | $0.072 \pm 0.013$ | $0.689 \pm 0.013$ |
| | VAE (MTL) | $0.045 \pm 0.007$ | $0.158 \pm 0.010$ | $0.098 \pm 0.008$ | $0.693 \pm 0.000$ |
| | GRAPE (MTL) | $0.044 \pm 0.011$ | $0.150 \pm 0.024$ | $0.157 \pm 0.058$ | $0.617 \pm 0.065$ |
| | PIG | $\mathbf{0.027} \pm \mathbf{0.005}$ | $\mathbf{0.118} \pm \mathbf{0.009}$ | $\mathbf{0.004} \pm \mathbf{0.002}$ | $\mathbf{0.100} \pm \mathbf{0.013}$ |
| Diabetes | SI | $0.146 \pm 0.018$ | $0.315 \pm 0.02$ | $0.207 \pm 0.029$ | - |
| | MICE | $0.205 \pm 0.008$ | $0.364 \pm 0.008$ | $0.250 \pm 0.032$ | - |
| | MissForest | $0.214 \pm 0.006$ | $0.383 \pm 0.001$ | $0.173 \pm 0.014$ | - |
| | AE | $0.144 \pm 0.033$ | $0.309 \pm 0.042$ | $0.214 \pm 0.026$ | - |
| | VAE | $0.155 \pm 0.026$ | $0.325 \pm 0.031$ | $0.216 \pm 0.035$ | - |
| | GRAPE | $0.185 \pm 0.055$ | $0.332 \pm 0.050$ | $0.605 \pm 0.408$ | - |
| | GAIN | $\mathbf{0.047} \pm \mathbf{0.006}$ | $\mathbf{0.146} \pm 0.008$ | $\mathbf{0.014} \pm \mathbf{0.001}$ | - |
| | MIRACLE | $1.071 \pm 0.128$ | $0.792 \pm 0.053$ | $13.457 \pm 0.135$ | - |
| | HyperImpute | $0.206 \pm 0.013$ | $0.375 \pm 0.017$ | $0.132 \pm 0.016$ | - |
| | AE (MTL) | $0.140 \pm 0.027$ | $0.308 \pm 0.028$ | $0.220 \pm 0.032$ | - |
| | VAE (MTL) | $0.141 \pm 0.029$ | $0.302 \pm 0.033$ | $0.227 \pm 0.034$ | - |
| | GRAPE (MTL) | $0.123 \pm 0.022$ | $0.285 \pm 0.028$ | $0.226 \pm 0.019$ | - |
| | PIG | $0.122 \pm 0.027$ | $0.280 \pm 0.035$ | $0.034 \pm 0.009$ | - |

## J EXPERIMENTAL RESULTS FOR SIX CLASSIFICATION DATASETS

Table 8: The classification and imputation performance for six datasets. Bold: best score. Underline: runner-up.

| Data | Model | Classification Task | | | | Imputation Task | |
|---|---|---|---|---|---|---|---|
| | | Accuracy | Recall | Precision | F1 score | MSE | CE |
| Spam | SI | 0.831 ± 0.025 | 0.805 ± 0.031 | 0.838 ± 0.022 | 0.814 ± 0.029 | 0.016 ± 0.003 | - |
| | MICE | 0.803 ± 0.007 | 0.783 ± 0.013 | 0.796 ± 0.006 | 0.787 ± 0.012 | 0.043 ± 0.003 | - |
| | MissForest | 0.850 ± 0.007 | 0.829 ± 0.010 | 0.856 ± 0.001 | 0.837 ± 0.009 | 0.014 ± 0.002 | - |
| | AE | 0.828 ± 0.020 | 0.795 ± 0.026 | 0.848 ± 0.016 | 0.807 ± 0.025 | 0.020 ± 0.003 | - |
| | VAE | 0.818 ± 0.032 | 0.783 ± 0.043 | 0.846 ± 0.021 | 0.793 ± 0.044 | 0.018 ± 0.003 | - |
| | GRAPE | 0.816 ± 0.063 | 0.805 ± 0.060 | 0.810 ± 0.061 | 0.807 ± 0.061 | 0.032 ± 0.015 | - |
| | GAIN | 0.858 ± 0.017 | 0.828 ± 0.020 | 0.878 ± 0.018 | 0.842 ± 0.020 | 0.019 ± 0.002 | - |
| | MIRACLE | 0.596 ± 0.001 | 0.581 ± 0.000 | 0.579 ± 0.001 | 0.580 ± 0.001 | 7.029 ± 0.037 | - |
| | HyperImpute | 0.850 ± 0.007 | 0.830 ± 0.009 | 0.852 ± 0.006 | 0.838 ± 0.008 | 0.014 ± 0.001 | - |
| | AE (MTL) | 0.883 ± 0.011 | 0.866 ± 0.013 | 0.890 ± 0.010 | 0.874 ± 0.012 | 0.017 ± 0.007 | - |
| | VAE (MTL) | 0.883 ± 0.019 | 0.869 ± 0.024 | 0.887 ± 0.015 | 0.875 ± 0.021 | 0.017 ± 0.002 | - |
| | GRAPE (MTL) | 0.843 ± 0.025 | 0.814 ± 0.029 | 0.858 ± 0.022 | 0.825 ± 0.027 | 0.021 ± 0.003 | - |
| | PIG | **0.899 ± 0.020** | **0.887 ± 0.022** | **0.901 ± 0.020** | **0.892 ± 0.022** | **0.006 ± 0.001** | - |
| Mobile | SI | 0.590 ± 0.041 | 0.594 ± 0.038 | 0.588 ± 0.040 | 0.582 ± 0.041 | 0.334 ± 0.015 | 0.676 ± 0.012 |
| | MICE | 0.505 ± 0.015 | 0.514 ± 0.018 | 0.520 ± 0.016 | 0.495 ± 0.015 | 0.616 ± 0.002 | 0.699 ± 0.003 |
| | MissForest | 0.612 ± 0.025 | 0.620 ± 0.028 | 0.622 ± 0.030 | 0.601 ± 0.024 | 0.339 ± 0.002 | 0.707 ± 0.000 |
| | AE | 0.609 ± 0.047 | 0.610 ± 0.048 | 0.605 ± 0.053 | 0.601 ± 0.052 | 0.348 ± 0.015 | 0.676 ± 0.008 |
| | VAE | 0.616 ± 0.077 | 0.611 ± 0.070 | 0.606 ± 0.076 | 0.603 ± 0.077 | 0.336 ± 0.013 | 0.665 ± 0.009 |
| | GRAPE | 0.653 ± 0.034 | 0.658 ± 0.030 | 0.724 ± 0.038 | 0.649 ± 0.036 | 1.431 ± 0.231 | 0.727 ± 0.041 |
| | GAIN | 0.581 ± 0.025 | 0.579 ± 0.020 | 0.573 ± 0.016 | 0.566 ± 0.026 | 0.397 ± 0.016 | 0.717 ± 0.001 |
| | MIRACLE | 0.367 ± 0.006 | 0.374 ± 0.001 | 0.468 ± 0.003 | 0.316 ± 0.002 | 14.271 ± 0.545 | 0.690 ± 0.000 |
| | HyperImpute | 0.582 ± 0.007 | 0.589 ± 0.010 | 0.584 ± 0.004 | 0.577 ± 0.011 | 0.343 ± 0.004 | 0.714 ± 0.009 |
| | AE (MTL) | 0.758 ± 0.036 | 0.760 ± 0.034 | 0.791 ± 0.030 | 0.763 ± 0.032 | 0.336 ± 0.009 | 0.695 ± 0.026 |
| | VAE (MTL) | 0.750 ± 0.021 | 0.750 ± 0.020 | 0.777 ± 0.022 | 0.754 ± 0.022 | 0.343 ± 0.015 | 0.727 ± 0.034 |
| | GRAPE (MTL) | 0.757 ± 0.033 | 0.756 ± 0.030 | 0.785 ± 0.034 | 0.758 ± 0.030 | 0.371 ± 0.006 | 0.697 ± 0.001 |
| | PIG | **0.952 ± 0.017** | **0.951 ± 0.018** | **0.952 ± 0.017** | **0.95 ± 0.018** | 0.084 ± 0.007 | **0.222 ± 0.042** |
| Pulsar | SI | 0.906 ± 0.026 | 0.904 ± 0.026 | 0.909 ± 0.026 | 0.905 ± 0.026 | 0.043 ± 0.006 | - |
| | MICE | 0.903 ± 0.007 | 0.897 ± 0.009 | 0.915 ± 0.003 | 0.901 ± 0.008 | 0.049 ± 0.006 | - |
| | MissForest | 0.891 ± 0.003 | 0.885 ± 0.006 | 0.902 ± 0.000 | 0.888 ± 0.004 | 0.020 ± 0.004 | - |
| | AE | 0.899 ± 0.022 | 0.899 ± 0.023 | 0.901 ± 0.021 | 0.899 ± 0.022 | 0.148 ± 0.027 | - |
| | VAE | 0.883 ± 0.026 | 0.885 ± 0.025 | 0.885 ± 0.025 | 0.882 ± 0.026 | 0.157 ± 0.017 | - |
| | GRAPE | 0.827 ± 0.048 | 0.829 ± 0.046 | 0.831 ± 0.047 | 0.827 ± 0.048 | 0.281 ± 0.084 | - |
| | GAIN | 0.863 ± 0.009 | 0.861 ± 0.009 | 0.880 ± 0.005 | 0.861 ± 0.009 | 0.053 ± 0.004 | - |
| | MIRACLE | 0.626 ± 0.028 | 0.635 ± 0.024 | 0.642 ± 0.024 | 0.623 ± 0.028 | 9.823 ± 0.136 | - |
| | HyperImpute | 0.849 ± 0.012 | 0.828 ± 0.015 | 0.852 ± 0.007 | 0.836 ± 0.014 | **0.014 ± 0.001** | - |
| | AE (MTL) | 0.910 ± 0.026 | 0.909 ± 0.025 | 0.911 ± 0.025 | 0.910 ± 0.026 | 0.114 ± 0.017 | - |
| | VAE (MTL) | 0.880 ± 0.012 | 0.880 ± 0.013 | 0.881 ± 0.011 | 0.879 ± 0.012 | 0.160 ± 0.016 | - |
| | GRAPE (MTL) | 0.912 ± 0.023 | 0.912 ± 0.023 | 0.916 ± 0.022 | 0.911 ± 0.024 | 0.164 ± 0.011 | - |
| | PIG | **0.923 ± 0.018** | **0.923 ± 0.018** | **0.926 ± 0.017** | **0.922 ± 0.018** | 0.046 ± 0.002 | - |
| Breast Cancer | SI | **0.962 ± 0.033** | 0.949 ± 0.047 | **0.970 ± 0.026** | 0.957 ± 0.040 | 0.021 ± 0.005 | - |
| | MICE | 0.771 ± 0.029 | 0.486 ± 0.046 | 0.453 ± 0.158 | 0.455 ± 0.068 | 0.071 ± 0.007 | - |
| | MissForest | 0.762 ± 0.000 | 0.474 ± 0.009 | 0.448 ± 0.144 | 0.432 ± 0.000 | 0.061 ± 0.009 | - |
| | AE | 0.943 ± 0.029 | 0.923 ± 0.041 | 0.959 ± 0.020 | 0.936 ± 0.034 | 0.086 ± 0.019 | - |
| | VAE | 0.945 ± 0.024 | 0.928 ± 0.035 | 0.959 ± 0.016 | 0.939 ± 0.029 | 0.088 ± 0.020 | - |
| | GRAPE | 0.888 ± 0.066 | 0.882 ± 0.057 | 0.899 ± 0.059 | 0.880 ± 0.065 | 0.501 ± 0.522 | - |
| | GAIN | 0.800 ± 0.029 | 0.500 ± 0.000 | 0.900 ± 0.014 | 0.444 ± 0.009 | 0.286 ± 0.046 | - |
| | MIRACLE | 0.752 ± 0.029 | 0.474 ± 0.009 | 0.443 ± 0.130 | 0.429 ± 0.010 | 9.879 ± 0.233 | - |
| | HyperImpute | 0.771 ± 0.029 | 0.474 ± 0.009 | 0.453 ± 0.159 | 0.435 ± 0.009 | 0.055 ± 0.019 | - |
| | AE (MTL) | 0.947 ± 0.021 | 0.932 ± 0.029 | 0.953 ± 0.020 | 0.939 ± 0.024 | 0.077 ± 0.011 | - |
| | VAE (MTL) | 0.943 ± 0.019 | 0.932 ± 0.025 | 0.946 ± 0.025 | 0.937 ± 0.022 | 0.102 ± 0.016 | - |
| | GRAPE (MTL) | 0.940 ± 0.036 | 0.912 ± 0.048 | 0.959 ± 0.024 | 0.928 ± 0.042 | 0.136 ± 0.027 | - |
| | PIG | **0.962 ± 0.029** | **0.956 ± 0.033** | 0.966 ± 0.025 | **0.960 ± 0.030** | **0.010 ± 0.006** | - |
| Faults | SI | 0.631 ± 0.022 | 0.609 ± 0.088 | 0.672 ± 0.040 | 0.591 ± 0.080 | 0.094 ± 0.012 | - |
| | MICE | 0.583 ± 0.009 | 0.457 ± 0.009 | 0.561 ± 0.017 | 0.465 ± 0.010 | 0.302 ± 0.010 | - |
| | MissForest | 0.617 ± 0.022 | 0.472 ± 0.022 | 0.721 ± 0.005 | 0.475 ± 0.023 | **0.079 ± 0.002** | - |
| | AE | 0.617 ± 0.032 | 0.534 ± 0.058 | 0.629 ± 0.095 | 0.553 ± 0.062 | 0.241 ± 0.054 | - |
| | VAE | 0.635 ± 0.035 | 0.558 ± 0.061 | 0.693 ± 0.062 | 0.576 ± 0.052 | 0.290 ± 0.104 | - |
| | GRAPE | 0.437 ± 0.087 | 0.435 ± 0.090 | 0.415 ± 0.098 | 0.378 ± 0.081 | 1.174 ± 0.625 | - |
| | GAIN | 0.612 ± 0.005 | 0.467 ± 0.006 | 0.616 ± 0.006 | 0.469 ± 0.000 | 0.079 ± 0.004 | - |
| | MIRACLE | 0.354 ± 0.000 | 0.287 ± 0.007 | 0.287 ± 0.006 | 0.262 ± 0.002 | 9.029 ± 0.193 | - |
| | HyperImpute | 0.606 ± 0.002 | 0.463 ± 0.016 | **0.723 ± 0.005** | 0.468 ± 0.017 | 0.085 ± 0.010 | - |
| | AE (MTL) | 0.525 ± 0.029 | 0.462 ± 0.050 | 0.550 ± 0.105 | 0.470 ± 0.055 | 0.200 ± 0.015 | - |
| | VAE (MTL) | 0.544 ± 0.033 | 0.465 ± 0.068 | 0.576 ± 0.087 | 0.477 ± 0.069 | 0.263 ± 0.010 | - |
| | GRAPE (MTL) | 0.565 ± 0.027 | 0.546 ± 0.055 | 0.547 ± 0.050 | 0.505 ± 0.044 | 0.513 ± 0.148 | - |
| | PIG | **0.692 ± 0.026** | **0.714 ± 0.048** | 0.690 ± 0.042 | **0.686 ± 0.042** | 0.085 ± 0.134 | - |
| Wine | SI | 0.554 ± 0.048 | 0.393 ± 0.101 | 0.759 ± 0.071 | 0.383 ± 0.105 | 0.067 ± 0.011 | - |
| | MICE | 0.543 ± 0.016 | 0.230 ± 0.045 | 0.697 ± 0.045 | 0.210 ± 0.045 | 0.061 ± 0.004 | - |
| | MissForest | 0.539 ± 0.026 | 0.213 ± 0.014 | 0.828 ± 0.040 | 0.192 ± 0.012 | 0.047 ± 0.006 | - |
| | AE | 0.541 ± 0.030 | 0.395 ± 0.108 | 0.800 ± 0.055 | 0.381 ± 0.109 | 0.080 ± 0.018 | - |
| | VAE | 0.561 ± 0.046 | **0.428 ± 0.150** | 0.797 ± 0.069 | **0.415 ± 0.152** | 0.086 ± 0.019 | - |
| | GRAPE | 0.511 ± 0.046 | 0.375 ± 0.111 | 0.606 ± 0.151 | 0.336 ± 0.123 | 0.286 ± 0.085 | - |
| | GAIN | 0.537 ± 0.008 | 0.233 ± 0.045 | **0.847 ± 0.007** | 0.207 ± 0.046 | 0.050 ± 0.001 | - |
| | MIRACLE | 0.491 ± 0.013 | 0.243 ± 0.043 | 0.807 ± 0.043 | 0.253 ± 0.043 | 9.623 ± 0.012 | - |
| | HyperImpute | 0.534 ± 0.010 | 0.227 ± 0.055 | 0.843 ± 0.005 | 0.207 ± 0.055 | 0.043 ± 0.002 | - |
| | AE (MTL) | **0.563 ± 0.042** | 0.290 ± 0.087 | 0.716 ± 0.084 | 0.282 ± 0.093 | 0.083 ± 0.008 | - |
| | VAE (MTL) | 0.558 ± 0.049 | 0.374 ± 0.111 | 0.758 ± 0.058 | 0.352 ± 0.089 | 0.084 ± 0.013 | - |
| | GRAPE (MTL) | 0.561 ± 0.029 | 0.358 ± 0.118 | 0.814 ± 0.071 | 0.337 ± 0.121 | 0.138 ± 0.050 | - |
| | PIG | 0.561 ± 0.020 | 0.405 ± 0.078 | 0.775 ± 0.037 | 0.397 ± 0.080 | **0.013 ± 0.003** | - |

# K EXPERIMENTAL RESULTS FOR SENSITIVITY ANALYSIS

We measured how sensitive each model is to missing rates. We varied the missing rates from 5% to 50% and reported the results. The sensitivity analysis was implemented using the "Mobile" dataset.

PIG showed remarkable robustness to various missing rates compared to baseline models. Here, we present the results of a sensitivity analysis for some representative methods, SI, GAIN, GRAPE, and PIG (Figure 10). While PIG also exhibits performance degradation as missing rates increase, the other methods quickly saturate to lower performance. Refer to Table 9 for more detail.

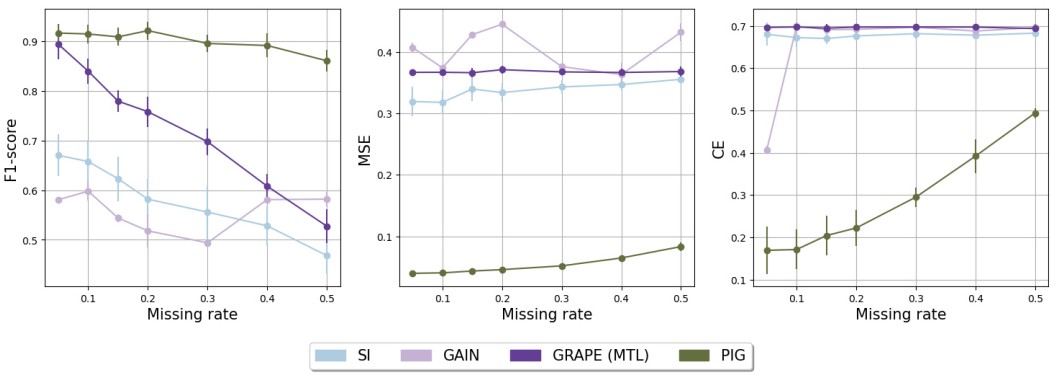

Figure 10: Sensitivity analysis on different missing rates. Left: classification performance (F1-score). Center: imputation performance (MSE) for numeric features. Right: imputation performance (CE) for categorical features.

Table 9: Sensitivity analysis to different missing rates. Bold: best score. Underline: runner-up.

| Missing rates | Model | Imputation | | Classification | | | |
|---|---|---|---|---|---|---|---|
| | | MSE | CE | Accuracy | Recall | Precision | F1-score |
| 0.05 | SI | 0.319 ± 0.024 | 0.680 ± 0.027 | 0.679 ± 0.041 | 0.678 ± 0.038 | 0.675 ± 0.041 | 0.670 ± 0.042 |
| | MICE | 0.622 ± 0.019 | 0.683 ± 0.012 | 0.532 ± 0.011 | 0.540 ± 0.013 | 0.548 ± 0.019 | 0.518 ± 0.011 |
| | MissForest | 0.319 ± 0.017 | 0.686 ± 0.015 | 0.559 ± 0.003 | 0.567 ± 0.005 | 0.564 ± 0.005 | 0.545 ± 0.004 |
| | AE | 0.344 ± 0.019 | 0.679 ± 0.019 | 0.656 ± 0.050 | 0.662 ± 0.048 | 0.658 ± 0.050 | 0.651 ± 0.050 |
| | VAE | 0.328 ± 0.019 | 0.668 ± 0.019 | 0.677 ± 0.047 | 0.678 ± 0.044 | 0.674 ± 0.044 | 0.667 ± 0.044 |
| | AE (MTL) | 0.339 ± 0.026 | 0.708 ± 0.050 | 0.892 ± 0.019 | 0.893 ± 0.018 | 0.892 ± 0.019 | 0.891 ± 0.019 |
| | VAE (MTL) | 0.349 ± 0.023 | 0.754 ± 0.043 | 0.893 ± 0.025 | 0.892 ± 0.024 | 0.896 ± 0.023 | 0.892 ± 0.024 |
| | GRAPE (MTL) | 0.367 ± 0.005 | 0.697 ± 0.005 | 0.897 ± 0.027 | 0.897 ± 0.030 | 0.902 ± 0.025 | 0.894 ± 0.030 |
| | GRAPE | 1.272 ± 0.386 | 0.696 ± 0.087 | 0.893 ± 0.020 | 0.891 ± 0.022 | 0.902 ± 0.017 | 0.892 ± 0.021 |
| | MIRACLE | 13.732 ± 0.030 | 0.691 ± 0.001 | 0.417 ± 0.024 | 0.424 ± 0.023 | 0.479 ± 0.027 | 0.394 ± 0.026 |
| | HyperImpute | 0.340 ± 0.005 | 0.686 ± 0.012 | 0.568 ± 0.007 | 0.577 ± 0.008 | 0.568 ± 0.008 | 0.556 ± 0.007 |
| | GAIN | 0.407 ± 0.008 | 0.407 ± 0.009 | 0.587 ± 0.005 | 0.588 ± 0.001 | 0.584 ± 0.001 | 0.581 ± 0.001 |
| | PIG | **0.040 ± 0.001** | **0.169 ± 0.056** | **0.917 ± 0.018** | **0.918 ± 0.016** | **0.923 ± 0.019** | **0.917 ± 0.018** |
| 0.10 | SI | 0.318 ± 0.019 | 0.673 ± 0.023 | 0.671 ± 0.042 | 0.669 ± 0.040 | 0.671 ± 0.040 | 0.658 ± 0.043 |
| | MICE | 0.613 ± 0.022 | 0.679 ± 0.006 | 0.564 ± 0.012 | 0.571 ± 0.006 | 0.578 ± 0.003 | 0.551 ± 0.012 |
| | MissForest | 0.343 ± 0.003 | 0.680 ± 0.001 | 0.578 ± 0.007 | 0.585 ± 0.009 | 0.584 ± 0.010 | 0.566 ± 0.005 |
| | AE | 0.344 ± 0.021 | 0.672 ± 0.019 | 0.642 ± 0.064 | 0.642 ± 0.064 | 0.637 ± 0.062 | 0.633 ± 0.064 |
| | VAE | 0.342 ± 0.016 | 0.672 ± 0.017 | 0.644 ± 0.050 | 0.649 ± 0.044 | 0.642 ± 0.052 | 0.637 ± 0.052 |
| | AE (MTL) | 0.340 ± 0.013 | 0.682 ± 0.042 | 0.853 ± 0.026 | 0.852 ± 0.026 | 0.860 ± 0.024 | 0.852 ± 0.026 |
| | VAE (MTL) | 0.339 ± 0.017 | 0.738 ± 0.036 | 0.831 ± 0.024 | 0.831 ± 0.027 | 0.840 ± 0.022 | 0.831 ± 0.025 |
| | GRAPE (MTL) | 0.367 ± 0.005 | 0.698 ± 0.001 | 0.841 ± 0.024 | 0.838 ± 0.025 | 0.853 ± 0.026 | 0.840 ± 0.025 |
| | GRAPE | 1.085 ± 0.418 | 0.714 ± 0.028 | 0.802 ± 0.037 | 0.803 ± 0.038 | 0.815 ± 0.034 | 0.799 ± 0.039 |
| | MIRACLE | 12.720 ± 0.521 | 0.691 ± 0.001 | 0.382 ± 0.008 | 0.391 ± 0.012 | 0.441 ± 0.017 | 0.335 ± 0.014 |
| | HyperImpute | 0.319 ± 0.001 | 0.676 ± 0.015 | 0.575 ± 0.002 | 0.581 ± 0.006 | 0.578 ± 0.008 | 0.565 ± 0.004 |
| | GAIN | 0.374 ± 0.002 | 0.700 ± 0.010 | 0.604 ± 0.019 | 0.605 ± 0.020 | 0.603 ± 0.020 | 0.598 ± 0.019 |
| | PIG | **0.041 ± 0.002** | **0.171 ± 0.047** | **0.916 ± 0.018** | **0.915 ± 0.018** | **0.921 ± 0.019** | **0.915 ± 0.019** |
| 0.15 | SI | 0.339 ± 0.020 | 0.670 ± 0.014 | 0.631 ± 0.043 | 0.633 ± 0.045 | 0.630 ± 0.046 | 0.623 ± 0.045 |
| | MICE | 0.737 ± 0.030 | 0.690 ± 0.016 | 0.506 ± 0.005 | 0.515 ± 0.001 | 0.504 ± 0.004 | 0.493 ± 0.008 |
| | MissForest | 0.329 ± 0.001 | 0.680 ± 0.006 | 0.576 ± 0.002 | 0.581 ± 0.004 | 0.584 ± 0.005 | 0.568 ± 0.000 |
| | AE | 0.336 ± 0.020 | 0.679 ± 0.007 | 0.629 ± 0.062 | 0.630 ± 0.056 | 0.625 ± 0.060 | 0.618 ± 0.061 |
| | VAE | 0.344 ± 0.016 | 0.673 ± 0.012 | 0.628 ± 0.052 | 0.637 ± 0.050 | 0.629 ± 0.052 | 0.622 ± 0.051 |
| | AE (MTL) | 0.342 ± 0.012 | 0.696 ± 0.020 | 0.796 ± 0.026 | 0.795 ± 0.022 | 0.809 ± 0.023 | 0.798 ± 0.022 |
| | VAE (MTL) | 0.347 ± 0.014 | 0.744 ± 0.040 | 0.799 ± 0.026 | 0.797 ± 0.026 | 0.818 ± 0.025 | 0.800 ± 0.026 |
| | GRAPE (MTL) | 0.366 ± 0.007 | 0.695 ± 0.010 | 0.779 ± 0.022 | 0.780 ± 0.022 | 0.797 ± 0.021 | 0.780 ± 0.021 |
| | GRAPE | 1.380 ± 0.250 | 0.739 ± 0.049 | 0.757 ± 0.052 | 0.757 ± 0.049 | 0.791 ± 0.045 | 0.755 ± 0.055 |
| | MIRACLE | 13.192 ± 0.363 | 0.692 ± 0.001 | 0.444 ± 0.003 | 0.447 ± 0.007 | 0.519 ± 0.007 | 0.406 ± 0.001 |
| | HyperImpute | 0.312 ± 0.011 | 0.312 ± 0.000 | 0.552 ± 0.007 | 0.560 ± 0.008 | 0.565 ± 0.002 | 0.545 ± 0.008 |
| | GAIN | 0.428 ± 0.002 | 0.691 ± 0.007 | 0.548 ± 0.010 | 0.553 ± 0.008 | 0.550 ± 0.011 | 0.544 ± 0.008 |
| | PIG | **0.043 ± 0.003** | **0.204 ± 0.047** | **0.910 ± 0.018** | **0.910 ± 0.018** | **0.912 ± 0.019** | **0.909 ± 0.018** |
| 0.20 | SI | 0.334 ± 0.015 | 0.676 ± 0.012 | 0.590 ± 0.041 | 0.594 ± 0.038 | 0.588 ± 0.040 | 0.582 ± 0.041 |
| | MICE | 0.561 ± 0.017 | 0.682 ± 0.005 | 0.567 ± 0.010 | 0.575 ± 0.011 | 0.569 ± 0.009 | 0.558 ± 0.011 |
| | MissForest | 0.352 ± 0.013 | 0.683 ± 0.009 | 0.558 ± 0.008 | 0.566 ± 0.006 | 0.567 ± 0.010 | 0.547 ± 0.004 |
| | AE | 0.348 ± 0.015 | 0.676 ± 0.008 | 0.609 ± 0.047 | 0.610 ± 0.048 | 0.605 ± 0.053 | 0.601 ± 0.052 |
| | VAE | 0.336 ± 0.013 | 0.665 ± 0.009 | 0.616 ± 0.077 | 0.611 ± 0.070 | 0.606 ± 0.076 | 0.603 ± 0.077 |
| | AE (MTL) | 0.336 ± 0.009 | 0.695 ± 0.026 | 0.758 ± 0.036 | 0.760 ± 0.034 | 0.791 ± 0.030 | 0.763 ± 0.032 |
| | VAE (MTL) | 0.343 ± 0.015 | 0.727 ± 0.034 | 0.750 ± 0.021 | 0.750 ± 0.020 | 0.777 ± 0.022 | 0.754 ± 0.022 |
| | GRAPE (MTL) | 0.371 ± 0.006 | 0.697 ± 0.001 | 0.757 ± 0.033 | 0.756 ± 0.030 | 0.785 ± 0.034 | 0.758 ± 0.030 |
| | GRAPE | 1.431 ± 0.231 | 0.727 ± 0.041 | 0.653 ± 0.034 | 0.658 ± 0.030 | 0.724 ± 0.038 | 0.649 ± 0.036 |
| | MIRACLE | 12.429 ± 0.107 | 0.691 ± 0.000 | 0.373 ± 0.006 | 0.377 ± 0.010 | 0.407 ± 0.004 | 0.327 ± 0.005 |
| | HyperImpute | 0.340 ± 0.004 | 0.677 ± 0.009 | 0.572 ± 0.009 | 0.578 ± 0.016 | 0.579 ± 0.019 | 0.562 ± 0.015 |
| | GAIN | 0.445 ± 0.004 | 0.692 ± 0.002 | 0.530 ± 0.032 | 0.523 ± 0.035 | 0.521 ± 0.036 | 0.518 ± 0.033 |
| | PIG | **0.046 ± 0.002** | **0.222 ± 0.042** | **0.923 ± 0.018** | **0.923 ± 0.018** | **0.926 ± 0.017** | **0.922 ± 0.018** |
| 0.30 | SI | 0.343 ± 0.011 | 0.682 ± 0.011 | 0.565 ± 0.052 | 0.565 ± 0.048 | 0.562 ± 0.048 | 0.556 ± 0.052 |
| | MICE | 0.630 ± 0.020 | 0.674 ± 0.005 | 0.521 ± 0.018 | 0.528 ± 0.015 | 0.518 ± 0.021 | 0.513 ± 0.018 |
| | MissForest | 0.330 ± 0.006 | 0.687 ± 0.006 | 0.526 ± 0.018 | 0.535 ± 0.015 | 0.513 ± 0.021 | 0.506 ± 0.023 |
| | AE | 0.335 ± 0.014 | 0.675 ± 0.009 | 0.570 ± 0.028 | 0.567 ± 0.028 | 0.565 ± 0.024 | 0.561 ± 0.026 |
| | VAE | 0.340 ± 0.010 | 0.677 ± 0.006 | 0.576 ± 0.036 | 0.575 ± 0.039 | 0.571 ± 0.043 | 0.570 ± 0.038 |
| | AE (MTL) | 0.339 ± 0.004 | 0.705 ± 0.023 | 0.679 ± 0.026 | 0.677 ± 0.028 | 0.731 ± 0.024 | 0.685 ± 0.026 |
| | VAE (MTL) | 0.345 ± 0.010 | 0.716 ± 0.034 | 0.667 ± 0.024 | 0.669 ± 0.022 | 0.734 ± 0.025 | 0.677 ± 0.022 |
| | GRAPE (MTL) | 0.367 ± 0.005 | 0.698 ± 0.001 | 0.694 ± 0.024 | 0.695 ± 0.024 | 0.745 ± 0.022 | 0.698 ± 0.027 |
| | GRAPE | 1.434 ± 0.269 | 0.741 ± 0.040 | 0.562 ± 0.047 | 0.562 ± 0.049 | 0.668 ± 0.065 | 0.552 ± 0.056 |
| | MIRACLE | 14.066 ± 0.116 | 0.692 ± 0.000 | 0.409 ± 0.001 | 0.416 ± 0.003 | 0.473 ± 0.008 | 0.391 ± 0.003 |
| | HyperImpute | 0.336 ± 0.006 | 0.680 ± 0.009 | 0.560 ± 0.002 | 0.568 ± 0.005 | 0.550 ± 0.002 | 0.540 ± 0.002 |
| | GAIN | 0.376 ± 0.001 | 0.697 ± 0.008 | 0.504 ± 0.003 | 0.507 ± 0.001 | 0.493 ± 0.000 | 0.494 ± 0.001 |
| | PIG | **0.052 ± 0.003** | **0.295 ± 0.023** | **0.897 ± 0.018** | **0.897 ± 0.018** | **0.899 ± 0.017** | **0.896 ± 0.018** |
| 0.40 | SI | 0.347 ± 0.009 | 0.678 ± 0.009 | 0.534 ± 0.036 | 0.531 ± 0.036 | 0.536 ± 0.041 | 0.528 ± 0.039 |
| | MICE | 0.611 ± 0.001 | 0.682 ± 0.004 | 0.549 ± 0.002 | 0.558 ± 0.002 | 0.546 ± 0.004 | 0.540 ± 0.003 |
| | MissForest | 0.363 ± 0.009 | 0.687 ± 0.011 | 0.564 ± 0.004 | 0.573 ± 0.008 | 0.571 ± 0.004 | 0.552 ± 0.004 |
| | AE | 0.337 ± 0.009 | 0.674 ± 0.005 | 0.520 ± 0.046 | 0.522 ± 0.044 | 0.530 ± 0.049 | 0.522 ± 0.045 |
| | VAE | 0.339 ± 0.008 | 0.672 ± 0.006 | 0.522 ± 0.042 | 0.522 ± 0.037 | 0.523 ± 0.044 | 0.518 ± 0.042 |
| | AE (MTL) | 0.337 ± 0.014 | 0.712 ± 0.015 | 0.618 ± 0.034 | 0.616 ± 0.033 | 0.699 ± 0.024 | 0.627 ± 0.033 |
| | VAE (MTL) | 0.344 ± 0.012 | 0.734 ± 0.036 | 0.606 ± 0.037 | 0.606 ± 0.039 | 0.680 ± 0.036 | 0.615 ± 0.036 |
| | GRAPE (MTL) | 0.366 ± 0.007 | 0.697 ± 0.001 | 0.600 ± 0.024 | 0.599 ± 0.022 | 0.690 ± 0.040 | 0.608 ± 0.025 |
| | GRAPE | 1.220 ± 0.342 | 0.722 ± 0.033 | 0.507 ± 0.070 | 0.517 ± 0.066 | 0.617 ± 0.060 | 0.487 ± 0.081 |
| | MIRACLE | 13.896 ± 0.318 | 0.692 ± 0.000 | 0.383 ± 0.004 | 0.386 ± 0.003 | 0.479 ± 0.007 | 0.361 ± 0.002 |
| | HyperImpute | 0.336 ± 0.008 | **0.336 ± 0.007** | 0.627 ± 0.009 | 0.633 ± 0.021 | 0.641 ± 0.023 | 0.614 ± 0.017 |
| | GAIN | 0.363 ± 0.019 | 0.688 ± 0.002 | 0.585 ± 0.000 | 0.585 ± 0.000 | 0.586 ± 0.003 | 0.581 ± 0.002 |
| | PIG | **0.065 ± 0.003** | 0.392 ± 0.040 | **0.894 ± 0.022** | **0.892 ± 0.025** | **0.895 ± 0.022** | **0.892 ± 0.024** |
| 0.50 | SI | 0.355 ± 0.006 | 0.683 ± 0.005 | 0.473 ± 0.032 | 0.476 ± 0.033 | 0.481 ± 0.041 | 0.468 ± 0.037 |
| | MICE | 0.594 ± 0.008 | 0.683 ± 0.009 | 0.536 ± 0.018 | 0.542 ± 0.014 | 0.548 ± 0.012 | 0.528 ± 0.016 |
| | MissForest | 0.356 ± 0.006 | 0.686 ± 0.013 | 0.515 ± 0.015 | 0.522 ± 0.011 | 0.517 ± 0.010 | 0.500 ± 0.015 |
| | AE | 0.343 ± 0.005 | 0.670 ± 0.005 | 0.470 ± 0.026 | 0.471 ± 0.026 | 0.495 ± 0.036 | 0.472 ± 0.028 |
| | VAE | 0.336 ± 0.006 | 0.670 ± 0.008 | 0.469 ± 0.028 | 0.468 ± 0.027 | 0.483 ± 0.034 | 0.465 ± 0.030 |
| | AE (MTL) | 0.337 ± 0.007 | 0.714 ± 0.024 | 0.534 ± 0.028 | 0.533 ± 0.025 | 0.649 ± 0.032 | 0.547 ± 0.027 |
| | VAE (MTL) | 0.342 ± 0.007 | 0.720 ± 0.021 | 0.533 ± 0.024 | 0.528 ± 0.027 | 0.639 ± 0.024 | 0.536 ± 0.033 |
| | GRAPE (MTL) | 0.368 ± 0.008 | 0.694 ± 0.010 | 0.521 ± 0.038 | 0.525 ± 0.031 | 0.639 ± 0.023 | 0.528 ± 0.034 |

| Missing rates | Model | Imputation | | Classification | | | |
| --- | --- | --- | --- | --- | --- | --- | --- |
| | | MSE | CE | Accuracy | Recall | Precision | F1-score |
| | GRAPE | $1.011 \pm 0.467$ | $0.705 \pm 0.031$ | $0.474 \pm 0.061$ | $0.478 \pm 0.049$ | $0.598 \pm 0.045$ | $0.453 \pm 0.071$ |
| | MIRACLE | $13.425 \pm 0.087$ | $0.691 \pm 0.001$ | $0.361 \pm 0.018$ | $0.366 \pm 0.012$ | $0.412 \pm 0.015$ | $0.330 \pm 0.012$ |
| | HyperImpute | $\underline{0.323 \pm 0.001}$ | $0.684 \pm 0.015$ | $0.549 \pm 0.002$ | $0.557 \pm 0.006$ | $0.555 \pm 0.008$ | $0.539 \pm 0.004$ |
| | GAIN | $0.432 \pm 0.015$ | $0.696 \pm 0.007$ | $\underline{0.599 \pm 0.014}$ | $\underline{0.600 \pm 0.014}$ | $\underline{0.598 \pm 0.018}$ | $\underline{0.582 \pm 0.015}$ |
| | PIG | $\mathbf{0.083 \pm 0.007}$ | $\mathbf{0.494 \pm 0.012}$ | $\mathbf{0.863 \pm 0.021}$ | $\mathbf{0.862 \pm 0.022}$ | $\mathbf{0.864 \pm 0.020}$ | $\mathbf{0.861 \pm 0.021}$ |

