# OpenReview forum: "Enhanced multi-task learning of imputation and prediction via feature relationship graph learning"
_ICLR.cc/2025/Conference — Submitted to ICLR 2025_

### Official Review · Reviewer_FDip · 2024-10-29

**Soundness:** 3
**Presentation:** 3
**Contribution:** 2
**Rating:** 5
**Confidence:** 4

**Summary:**

The authors assert that previous works did not take downstream tasks into account when imputing missing values and were vulnerable to irrelevant features, which could lead to overfitting.
To address this, they propose a multi-task learning approach called Prediction and Imputation via feature relationship Graph learning (PIG). This model leverages both imputation and downstream task losses to optimize performance, guided by both tasks.
Additionally, this model learns an adjacency matrix to capture relationships between features and incorporates this relational information for both imputation and the downstream task.
Through extensive experiments on both regression and classification datasets, the authors demonstrate the effectiveness of PIG.

**Strengths:**

- The proposed model achieves state-of-the-art performance with a simple and effective approach.
- The authors propose a novel approach to learning relationships between features, utilizing prior knowledge derived from feature importance in a decision tree.
- The authors conducted extensive experiments across various datasets and compared the model with multiple appropriate baseline methods to demonstrate the effectiveness of PIG.
- The paper is written to be easily understood.

**Weaknesses:**

1) The analysis of individual components is insufficient.
- To assess the contributions of each loss and examine the synergy between imputation and downstream task loss, the authors should conduct ablation studies for each loss.
- In particular, a detailed analysis of the contribution of the prior adjacency is needed to determine whether the performance of PIG is due to this beneficial prior information or the effective relational learning capability of GLL. For a deeper understanding, the authors could compare performance when using only relationships derived from prior knowledge.

2) Concerns remain regarding the fairness of the comparison.
- I conjecture that constructing a complement matrix with other imputation tools and using it as a starting point could create an ensemble effect. Since other baseline methods could also leverage this complement matrix as a starting point, it should be incorporated into the baseline models for a fair comparison.

**Questions:**

In the experimental section (lines 344–346), the authors state that 'MTL versions of existing models are implemented by modifying each method to learn imputation and prediction simultaneously as in PIG'.
Are they implying a two-step training strategy where the model is initially trained on the downstream task without masking, focusing only on optimizing the embedded part and predictor, and then subsequently updating all components together?
If not, I suggest using the same MTL scheme to ensure a fair comparison between baseline methods.

I will update my score if these concerns are all addressed.

---

### Official Review · Reviewer_RcJo · 2024-11-01

**Soundness:** 2
**Presentation:** 2
**Contribution:** 2
**Rating:** 3
**Confidence:** 4

**Summary:**

The paper introduces a model called PIG (Prediction and Imputation via feature-relationship Graph learning) aimed at improving both imputation and predictive tasks in datasets with missing values. Traditional imputation methods often ignore feature interdependencies, affecting downstream predictive accuracy.
To address this, PIG uses a graph-based multi-task learning approach that explicitly captures feature relationships. The model incorporates a Graph Learning Layer (GLL) to identify dependencies among features, which a Graph Residual Imputation Layer (GRIL) then uses to perform more accurate imputation by focusing on relevant features. Following this, the Prediction Layer (PL) leverages the imputed data for classification or regression tasks. Through a carefully designed training process involving pre-training and fine-tuning, PIG achieves improved accuracy and robustness, outperforming baseline models across various benchmarks.

**Strengths:**

The paper considers feature interdependence in handling missing data and designs a multi-task loss function. It employs two distinct modules—the Graph Learning Layer and the Graph Residual Imputation Layer—to identify interdependence relationships and use these relationships for imputation.

**Weaknesses:**

The paper lacks novelty, as the algorithm design is highly similar to MIRACLE, with both methods employing a loss function for multi-task learning according to feature interdependence. The main difference lies in the addition of multi-task learning on graphs through GNN (GLL and GRIL), but this innovation does not stand out significantly. Furthermore, there is no theoretical explanation to support the advantages of the proposed method in learning interdependence.

**Questions:**

1.	Regarding the lack of a full bar plot in the paper, could you provide a more detailed visualization?
2.	The datasets shown in the main text are primarily low-dimensional. How does the model perform on higher-dimensional datasets? If possible, please provide results.
3.	How does your approach handle different missing mechanisms?

---

### Official Review · Reviewer_gN1n · 2024-11-04

**Soundness:** 2
**Presentation:** 2
**Contribution:** 2
**Rating:** 5
**Confidence:** 4

**Summary:**

This paper addresses the limitations of existing missing value imputation methods, which have largely overlooked the relationship between features and downstream tasks, by proposing a novel approach called PIG. PIG integrates imputation and prediction tasks to improve overall performance.

First, using only non-missing values, it models feature interdependencies through Graph Construction (i.e., Graph Learning Layer; GLL). This GLL is then trained by comparing it to prior relationships derived from a decision tree algorithm by predicting a column using other columns. Additionally, a prediction layer (PL) is trained, completing the pre-training stage.

In the next phase, inputs with missing values are initially imputed using a simple method (Soft-Impute). The model then fine-tunes the GLL, Graph layer with residual connections (GRIL), and the PL, with the PL tailored to each specific downstream task, such as classification or regression.

**Strengths:**

* The proposed method effectively integrates imputation with downstream analysis, offering a unified framework that addresses both imputation and prediction tasks simultaneously.

* It demonstrates broad applicability by handling both numerical and categorical variables and by supporting both regression and classification tasks.

* Extensive experiments showcase the model's superior performance, underscoring its effectiveness across various scenarios.

**Weaknesses:**

* Limited technical novelty: Although new components are introduced, they are derived from general techniques, such as standard graph construction (GLL) and a basic graph layer with residual connections (GRIL).

* The model has a relatively complex training structure due to the two-phase training, initial imputation, and prior feature relationship modeling, which likely results in higher overall complexity compared to existing models.

* If feature interdependency is learned through the model, it should not only differ from prior relationships obtained via a Decision Tree but also capture meaningful relationships that the prior approach may have missed. It is essential to demonstrate the qualitative advantages of the model beyond quantitative results through additional experiments.

**Questions:**

* Could an analysis of the overall complexity of the training process be added?

* Can the effectiveness of pre-training for GLL and PL be evaluated separately to determine which has a greater impact? In the ablation study, the effect of pre-training only appears in the downstream task. Is pre-training for GLL truly necessary?

* Can you demonstrate experimentally that the model captures meaningful feature relationships beyond the performance improvement?

---

### Meta-Review · Area_Chair_4xaW · 2024-12-18

**Metareview:**

This paper introduces PIG, a model that integrates imputation and predictive tasks (classification or regression) using a graph-based approach to address missing values in datasets. Unlike traditional imputation methods, PIG explicitly captures feature interdependencies through a Graph Learning Layer (GLL) and uses a Graph Residual Imputation Layer (GRIL) to improve imputation accuracy. A Prediction Layer (PL) utilizes the imputed data for downstream tasks. The paper has some strengths like unified multi-task framework, strong empirical performance, and robustness to missing data.  However, it also has some relevant weaknesses such as limited novelty, a complex training process, and insufficient individual component analysis. Overall I believe that the paper requires a deep revision process to address these concerns.

**Additional Comments On Reviewer Discussion:**

The authors did not respond to the reviewers.

---

### Decision · Program_Chairs · 2025-01-22

Reject